



# Controls on the rates and products of particle attrition by bed-load collisions

Kimberly Litwin Miller[1,2] and Douglas Jerolmack[2]

[1]California Institute of Technology
[2]University of Pennsylvania

**Correspondence:** Kimberly Litwin Miller (kmiller2@caltech.edu)

**Abstract.** River rocks round through the process of impact attrition, where energetic collisions during bed-load transport induce chipping of the grain surface. This process is also important for bedrock erosion. Although previous work has shown that impact energy, lithology and shape are controlling factors for attrition rates, the functional dependence among these quantities is not settled. Here we examine these factors using a double-pendulum apparatus, that generates controlled collisions between two grains under conditions relevant for bed-load transport. We also determine the grain-size distributions (GSDs) of the attrition products. Two experimental results appear to support previous treatments of impact erosion as brittle fracture in the purely elastic regime: (i) mass loss is proportional to kinetic energy, and this proportionality is a function of previously identified material properties; and (ii) attrition product GSDs are well described by a Weibull distribution. Other observations, however, including the development of shallow and surface-parallel cracks, indicate that the common fatigue failure model is inappropriate. Rather, we propose that Hertzian fracture is the dominant mechanism that distinguishes chipping from fragmentation. We also identify an initial phase of rapid mass loss in which attrition is independent of energy and material properties; this is a shape effect associated with removal of very sharp corners. The apparent universality of both mass loss curves and attrition-product GSDs requires further investigation. Nonetheless, these findings are useful for interpreting the contribution of in-stream attrition to downstream fining and the production of sand, resulting from bed-load transport of river pebbles.

## 1 Introduction

Traveling downstream in a typical river, one observes river sediments becoming rounder in shape (Sneed and Folk, 1958; Adams, 1978) and smaller in size (Sternberg, 1875; Ferguson et al., 1996). While there is a debate over whether mechanical breakdown by impact attrition, or hydraulic sorting caused by relative transport rates, is responsible for fining patterns (Kodama, 1991; Ferguson et al., 1996; Gasparini et al., 1999; Lewin and Brewer, 2002), it is generally agreed that impact attrition is the chief mechanism producing the rounding of sediments (Kuenen, 1956; Sneed and Folk, 1958; Schumm and Stevens, 1973; Litwin Miller et al., 2014; Szabó et al., 2015; Novak-Szabo et al., 2018). Attrition is often called "abrasion" in the geological literature, but that term is avoided here since it has a mechanically distinct meaning in the comminution literature (Szabó et al., 2015). It is the process whereby river sediments are worn away due to energetic collisions with other grains and the channel bed during transport (Kuenen, 1956; Kodama, 1994b). Although there has been a great deal of previous work investigating



the process (Kodama, 1994b; Lewin and Brewer, 2002; Attal and Lave, 2009; Szabo et al., 2013; Litwin Miller et al., 2014; Szabó et al., 2015; Novak-Szabo et al., 2018), there is a lack of understanding of the fundamental physics involved in sediment impact attrition. Impact attrition by saltating bed-load particles is also a significant, and in many cases dominant, contributor to the erosion of bedrock river channels (Sklar and Dietrich, 1998, 2004). Similar to pebbles themselves, bedrock channels are often smoothed at small lengthscales by impact attrition (Wilson et al., 2013; Beer et al., 2017).

Sternberg (1875) attributed the downstream fining of grain size in rivers to impact attrition, and quantitatively described it with the exponential function:

$$D(x) = D_0 e^{-\alpha x} \qquad (1)$$

where $D(x)$ is the grain size at downstream distance $x$, $D_0$ is the initial grain size at $x = 0$, and $\alpha$ is the empirically determined diminution coefficient. Despite the fact that this expression lacks a mechanistic framework, $\alpha$ values remain the most commonly
applied method describing attrition rates. Most previous work on impact attrition has been through laboratory experiments (Krumbein, 1941; Kuenen, 1956; Kodama, 1994b; Lewin and Brewer, 2002; Attal and Lave, 2009) because of the difficulty in directly observing attrition in the field (Sneed and Folk, 1958; Kodama, 1994a). These experiments utilize tumbling mills or circular flumes to simulate bed-load transport; results are extrapolated to the field using the duration of the experiment as a proxy for downstream distance (Wentworth, 1919; Krumbein, 1941; Kuenen, 1956; Kodama, 1994b; Lewin and Brewer, 2002;
Attal and Lave, 2009). However, laboratory derived values of $\alpha$ (Wentworth, 1919; Krumbein, 1941; Kuenen, 1956; Kodama, 1994b; Lewin and Brewer, 2002; Attal and Lave, 2009) tend to be lower than those measured in the field (Ferguson et al., 1996; Hoey and Bluck, 1999; Morris and Williams, 1999). It has been suggested that this is because: impact energies in experiments are not as high as in the field (Kodama, 1994b); the assumption that experimental duration as a proxy for travel distance does not account for attrition in place (Schumm and Stevens, 1973); or added effects of hydraulic sorting on fining rates in the
field are not accounted for (Ferguson et al., 1996; Paola et al., 1992). These reasons highlight the need for a more mechanistic approach to the impact attrition process. Various studies have established that there is a linear relation between collision energy $\Delta E$ and the mass removed $\Delta M$ by the impact,

$$\Delta M = A \Delta E \qquad (2)$$

where $A$ is a parameter that collects relevant material properties (Bitter, 1963; Anderson, 1986; Kafui and Thornton, 1993;
Ghadiri and Zhang, 2002a; Le Bouteiller and Naaim, 2011; Wang et al., 2011). Indeed, Sternberg's law can be derived from this linear relation (Szabó et al., 2015; Novak-Szabo et al., 2018). There is less understanding and agreement, however, on what controls $A$. For commonly occurring rocks, different lithologies can lead to a difference in attrition rates of two orders of magnitude for the same collision energy (Attal and Lave, 2009). Based on the premise that rocks fail in tension under impact (Johnson, 1972), Sklar and Dietrich (Sklar and Dietrich, 2001, 2004) examined the relation between impact attrition rate and
tensile strength, finding an inverse square dependence. Bed-load transport experiments by Attal and Lave (2009) confirm that lithologies with low tensile strength, like weak sandstone, erode faster than those with higher values, like limestone and quartzite. Because rocks are very brittle, there is a long lineage of modeling impact attrition as purely elastic deformation



(e.g, (Shipway and Hutchings, 1993)). For energies below the fragmentation threshold — the typical scenario for bed-load transport of natural pebbles — it is typically assumed that elastic deformation slowly grows subsurface cracks until they merge

to produce products (Bitter, 1963). Besides tensile strength, this mechanism would indicate that material density and Young's modulus are also important parameters (Wang et al., 2011; Le Bouteiller and Naaim, 2011). While the sediment transport experiments mentioned offer tentative support for the brittle fracture approach, none of these studies examined individual collisions. Single-collision impact studies on very brittle materials (like glass) at high speeds suggest that fragmentation by impact attrition is similar to compression tests, though the peak stress must be modified (Shipway and Hutchings, 1993). Impact

studies at lower energies relevant for bed-load, however, typically show neither explosive nor fatigue-failure fragmentation. Rather, individual collisions produce a shower of small chips over a limited skin depth of the material. For very brittle ceramics and glasses, Hertzian fracture cones of shattered material form where the elastic shock wave from collision exceeds yield (Wilshaw, 1971; Rhee, 2001; Mohajerani and Spelt, 2010; Wang et al., 2017). This fracture process has been implicated in bedrock erosion by aeolian impacts (Greeley and Iversen, 1987). For semi-brittle materials, some ductile deformation occurs

resulting in detachment of finite-size chips due to flow (Ghadiri and Zhang, 2002a; Antonyuk et al., 2006). Even brittle materials likely experience some plastic deformation (Rhee, 2001; Momber, 2004b). Regardless, chipping implies that the more relevant material properties are those associated with lateral crack formation at the surface, rather than activation of cracks in the bulk (Greeley and Iversen, 1987; Momber, 2004a, b).

Most studies on impact attrition neglect the fine particles, or attrition products, produced from the process — even though

it has been hypothesized that these products heavily contribute to sand and silt populations found in rivers (Jerolmack and Brzinski, 2010). Experiments and theory examining the geometric evolution of pebbles during chipping predict that sediment can lose up to half of its original mass just from rounding the edges of an initially angular pebble (Domokos et al., 2014). With the large quantity of fines produced from chipping, it is necessary to understand the size distribution of these particles to understand their role in the river system. Kok (2011) found that the grain size distribution of dust aggregates follows a Weibull

distribution, in agreement with brittle fracture theory. The products of impact attrition for natural rocks, under collision energies relevant for bed load, have never been examined in this manner.

This paper explicitly isolates and investigates how lithology, shape and collision energy influence rates of impact attrition for particles and energies representative of pebble transport in rivers. First we determine how attrition rates scale with energy by performing well-controlled binary collision experiments. We conduct experiments on samples of different lithologies to

determine which measured material properties control the magnitude of attrition rates. Finally, we characterize the grain size distribution of the products created during the attrition process to determine whether it follows the expected Weibull distribution. Building on previous findings from bed-load attrition studies, this work considers the mechanics of fracture and damage in solid materials to provide a better understanding of the underlying physics.





## 2 Methods

### 2.1 Hypothesis and Experimental Approach

We have two hypotheses that guide our experimental design. First, we hypothesize that kinetic energy and lithology control attrition rates of river sediments. If we assume that rock deformation is completely elastic, and hence that rocks are purely brittle, then from mechanical considerations (Sklar and Dietrich, 2004; Attal and Lave, 2009; Le Bouteiller and Naaim, 2011; Wang et al., 2011) we can state that:

$$\Delta M = f(\Delta E, \rho, Y, \sigma), \tag{3}$$

where $\Delta M$ is the mass removed from an object after impact of energy $\Delta E$, and $\rho$, $Y$, and $\sigma$ are the material properties density, Young's modulus, and tensile strength, respectively. Dimensional analysis yields two dimensionless groups, $\Pi_1 = \frac{\sigma \Delta M}{\rho \Delta E}$ and $\Pi_2 = \frac{Y}{\sigma}$. Rewriting to solve for mass loss per unit impact energy, we obtain $\frac{\Delta M}{\Delta E} = f(A_b)$ where

$$A_b \equiv \frac{\rho Y}{\sigma^2}. \tag{4}$$

An alternative hypothesis is that impulsive collisions excite localized plastic deformation at the surface, similar to what is observed in micro/nano-indenter tests, and that $\frac{\Delta M}{\Delta E} = f(A_s)$ where

$$A_s \equiv \frac{\rho D H}{K_c^2}, \tag{5}$$

and $H$, $K_c$ and $D$ are hardness, fracture toughness and sample diameter, respectively, and the subscript $s$ denotes semi brittle Ghadiri and Zhang (2002b). We note that the ratio $H/K_c$ has been denoted the 'brittleness index', and shown to delineate the transition from purely elastic to elastic-plastic (or semi-brittle) deformation in natural rocks (Momber, 2004a). The utility of $A_b$ and $A_s$ for determining mass loss from attrition will be tested experimentally in this study.

The second hypothesis that guides this work is regarding the products of attrition. By the assumption from Griffith's fracture theory that pre-existing flaws are distributed independently within a material and activate randomly during a fracture event, it is expected that fragments produced follow a Weibull distribution (Gilvarry, 1961):

$$\frac{dN_f}{d \ln D_f} \propto D_f^{-2} \tag{6}$$

where $N_f$ is the number of fragments of size $D_f$. Kok (2011) discusses how this power-law relation follows from brittle fracture theory and is a consequence of the manner in which cracks nucleate and propagate within the material as stress is applied. These principles describe the full fragmentation of materials, meaning that the aggregate breaks into many small fragments with the largest fragment being significantly smaller than the parent particle. This is clearly not the case for chipping, in which the largest attrition product is much smaller than the parent. We will test whether the products of our attrition experiments follow the same power-law scaling for chipping. We hypothesize that brittle fragmentation may still occur, though over a small penetration depth near the impact site (Antonyuk et al., 2006).





## 2.2 Experimental Design and Methods

To simulate attrition between grains during saltation, while isolating the effects of impact energy on mass attrition, we examine
the amount of mass lost during a single collision event between two grains. Although collisions in water can be viscously
damped, for sufficiently large grains ($> 10^{-2}$ m) these collisions are semi-elastic and independent of the fluid (Schmeeckle
et al., 2001). Therefore, since our main goal is to determine the energy scaling of impact attrition, we conduct the experiments
in air instead of water for simplicity. The impact energies applied are comparable to those observed in nature. Experiments
are conducted using a "Newton's-cradle" style double pendulum housed within a transparent tank to allow for the collection
of the products of attrition (Fig. 1). Rock samples are attached to threaded rods within the tank by gluing flat-faced nuts to the
top of each sample. The rod with the impacting grain is lifted by a motor and then released once it reaches a desired height,
colliding with the stationary target grain. After the collision, a braking system steadies the target grain while the motor lifts
the impacting grain again for the next collision. Both rods containing impacting and target grains are able to rotate freely in
either direction, allowing attrition to occur evenly around the entire rock sample. To test the randomness of the grain rotation,
we filmed approximately 450 collisions between two test grains, recording the location of impact on both the impacting and
target grains. The distribution of impact locations indicates that the collisions occur preferentially on high curvature regions
of the protruding corners, as expected from geometric chipping theory (Firey, 1974; Domokos et al., 2014) (Fig. 2), but are
otherwise uniformly distributed around each grain. Grains are collided for a set interval of impacts, which increases throughout
the experiment from 50 to 10,000. After each set of collisions, the masses of both the impacting and target grains are measured
using a microbalance to determine the amount of mass lost due to attrition.

In order to measure the impact energy, we recorded videos at the beginning of every set of collisions with a high-speed
camera shooting at 1000 frames per second, mounted below the transparent bottom of the tank. We captured 5-10 collisions
per set; in each video, we tracked the location of the impacting grain over approximately 40 frames (0.04 s) up to the time
of collision. The impact velocity is measured as the slope of a linear fit to plots of travel distance versus time. The average
velocity for all experiments was approximately 1 m/s. The kinetic energy at impact ($\Delta E$) is then calculated as $\Delta E = \frac{1}{2}m_i v^2$,
where $m_i$ is the mass of the impacting grain at the beginning of the set and $v$ is the average velocity measured from all videos
in that particular set. Energies for experimental runs ranged from 0.035-0.220 J.

We conducted binary collision experiments on the following different materials: brick, quartz diorite, sandstone, schist, and
a volcaniclastic rock (Fig. 3). The brick was selected as a test material for its homogeneous structure. We used standard red clay
builders bricks. Both the quartz diorite and volcaniclastic rocks were collected in the Luquillo Mountains in northeastern Puerto
Rico. The quartz diorite is Tertiary in age and originates from a batholith on the southern side of the Luquillo Mountains (Pike
et al., 2010). The volcaniclastic rock comprises most of the mountain and was formed in the late Cretaceous from marine-
deposited volcanic sediments (Pike et al., 2010). The sandstone is a Triassic reddish arkose of the Stockton formation in
southeast Pennsylvania (Olsen, 1980). The schist is Wissahickon schist from southeast Pennsylvania and is highly deformed
due to regional metamorphism during the lower Paleozoic (Weiss, 1949). The brick was tested multiple times with different





size of impacting samples to study the effect of increased impact energy on abrasion rate. Table 1 lists the different rock types and sample sizes for each experimental run.

To control for shape effects on attrition rates, we initially cut all grains into cubes. Throughout the experiments, we tracked changes in the shape of both impacting and target grains using a laser displacement sensor to scan a single surface contour

around the grain. Scans are made at the beginning of each set of collisions by holding the sensor in a fixed position, while the grain is rotated at a constant rate of 3 rpm. A single contour for each grain is made by averaging 1-kHz laser data from approximately seven full rotations. The distance data are then smoothed using a high pass filter at the noise floor, which was determined from the time series of the entire dataset. The peak local curvature at each corner was calculated from the second derivative of the measured contour. The peaks from all four corners were averaged to give a mean value of corner curvature.

We also use a second method to characterize the shape evolution of the grains. Litwin Miller et al. (2014) demonstrated that the curvature entropy is a monotonically increasing quantity indicating the rounding of grains from attrition due to chipping. We measure this quantity from the laser-scanned contours using the methods outlined in Litwin Miller et al. (2014). Shape data were only collected for two sets of brick samples, and a single set of quartz diorite and sandstone specimens.

The relevant material properties needed to estimate $A_b$ and $A_s$ were measured for each lithology used in the experiment,

although our characterization was not entirely successful (see below). All measurements were made on 50-mm diameter cores cut from $\sim 0.5$ m rocks collected in the field (except for the brick specimens). The density of each core was calculated by dividing measured mass by volume determined from triplicate caliper measurements of the diameter and length of the cores. The average density of each lithology was determined from 10-15 cores. Tensile strength was measured using an indirect method called the Brazilian tensile test. This test measures the peak load for each sample loaded in compression, at which

point the sample fails in tension. A stress was applied to each sample by placing it in a specially fabricated metal fixture with a thin stick of bamboo between the sample and the fixture on each side of the loading plane. The bamboo sticks ensured that the load was only applied to the parallel radial axes at the top and bottom of each sample. The fixture was then placed between two metal plates of a Versa-Loader, an apparatus used to apply a compressional load at a constant strain rate to the sample. As the sample fails, fractionation occurs parallel to the loading direction; the peak load at failure is recorded. The tensile strength

of each sample is computed using this value of peak load in addition to the dimensions of the sample, through the following expression: $\sigma_t = \frac{2F_p}{\pi l D}$, where $F_p$ is the peak force applied to the sample at failure, $l$ is the length of the sample and $D$ is its diameter (Vutukuri, 1974). In order to reduce the uncertainty of the value of the tensile strength, 10-15 measurements for each rock type were made and averaged. Elastic modulus, also known as Young's modulus, is the proportionality factor between applied stress and elastic strain. We estimated this quantity using an Olsen Resonance Tester (RT-1) and the methods prescribed

by the ASTM C215 standard. In summary, an accelerometer is attached to the flat face of one end of the core, while a force is applied to the other end by hitting it with a small hammer. The applied force sends a vibrational wave through the core while the accelerometer records the longitudinal fundamental frequency. The elastic modulus ($Y$) is then calculated using the expression: $Y = \beta M f^2$, where $\beta$ is a shape correction factor equal to $5.093 \frac{l}{D^2}$ for cylindrical cores, $M$ is the sample mass, and $f$ is the recorded fundamental frequency. As with the other material properties, 10-15 measurements were taken and averaged together

to get a mean value for each lithology. This test produced reliable values of elastic modulus for brick, schist, and sandstone, but



we were not able to perform this test on the quartz diorite or volcaniclastic rocks because the sample specimens were not long enough. Instead ranges of elastic modulus values from the literature were compiled indicating that values for quartz diorite span a range from 20 to 70 GPa (Hughes and Jones, 1950; Merriam et al., 1970; Pratt et al., 1972; Fletcher et al., 2006), and values for volcaniclastic rock span a range from 5 to 50 GPa (Carlson and Wilkens, 1983; Apuani et al., 2005; Frolova, 2008; Rotonda et al., 2010).

Hardness was calculated using a standard Vickers Hardness test (ISO 6507-1:2005(E)); a Knoop micro-indentor with a pyramidal diamond tip was used, and a 1000-g load was applied for 10 s. Samples were cut into 1-cm cubes, and faces were polished with progressively finer grit down to 9 $\mu$m; this polishing is necessary in order to visualize the small indentations created by the test. Images at 50x magnification allowed measurement of the diagonal lengths of the indent (typically 10s of microns). A minimum of six and maximum of 15 indentations were measured for each sample, depending on image quality and our ability to find indents in the microscope; these values were averaged for each sample. Fracture toughness is the energy absorbed by a material before cracking; it is often estimated in a standard uniaxial stress test by integrating the area of the stress-strain curve. Our strength test only measured peak stress, however, so we could not employ this method. Moreover, such a bulk measure of fracture toughness is not necessarily indicative of shallow surface cracking associated with chipping (Ghadiri and Zhang, 2002b). It has been proposed that the length of cracks developed at the corners during a micro-indenter test can be used to calculate fracture toughness; this method, however, is not without controversy (Ghadiri and Zhang, 2002b). We attempted to estimate fracture toughness using this method, but unfortunately our results were inconclusive; crack development varied widely within a sample and from sample to sample, and was sometimes undetectable. We can only then crudely assess fracture toughness based on scaling relations derived from simplifying assumptions and empirical relations — though there is some backing from atomistic calculations (Emmerich, 2007). For brittle materials it is widely reported that fracture strength scales with Young's modulus, $\sigma_f \approx Y/10$ (Yuan and Xi, 2011), which relates to fracture toughness (with some assumptions) as $K_c \approx \sigma_f \sqrt{r_c}$ where $r_c$ is the crack tip radius of curvature (Emmerich, 2007). This implies that we may estimate the semi-brittle Attrition Number as $A_s = \rho D H / (Y^2 r_c)$, neglecting prefactors. We have no method available to estimate $r_c$ in our experiments; in the absence of information we will assume its variation is negligible compared to $H$ and $Y$. Table 1 lists all the values of material properties used in this study.

To better understand the modes of mechanical failure in the colliding particles, we prepared polished thin sections and examined them using a scanning electron microscope (SEM). The thin sections were parallel to planes that were perpendicular to the impact surface (Fig. 4a) and are imaged between 200 to 6000 magnification using an FEI 600 Quanta FEG environmental scanning electron microscope. Images were taken progressively along the edge of the samples (Fig. 4b/c) and compared to images of the sample interior. We then quantified the length scale over which damage occurred by taking between 600 to 1000 measurements of the length of the most interior crack that can be continuously tracked to the surface from different locations around each grain.

Finally, following each set of collisions, the products of the attrition process were collected from the bottom of the tank. Although we attempted to collect all of the products, small dust particles (< 1 $\mu$m) were observed to settle outside the tank, so we only reliably collected grains larger than that size. Fines produced throughout the entire experimental run for each pair





of rocks were combined into one population for grain size analysis; i.e., we did not track the evolution of attrition products through repeated collisions. Because the attrition products span a wide range of sizes, to fully characterize the grain size distribution (GSD) we employed two methods. First, to describe the coarse grains, we wet sieve the attrition products into three size fractions: $< 0.5$ mm, 0.5 mm to 1.0 mm, and $> 1.0$ mm. The coarser two fractions are dried in an oven and subsequently

weighed to determine their contribution to the entire size distribution. The grain size of the attrition products finer than 0.5 mm is measured using the Beckman Coulter laser diffraction particle analyzer, which determines the volumetric GSD by deconstructing the diffraction pattern produced by shining a laser through a liquid solution containing the fine-grained sample. Because of the large quantity of fines produced in the experiments, we perform repeated subsampled measurements of grain size using the Coulter counter. We select five subsamples from a mixture of fine particles and deionized water. To ensure

consistent subsampling of a homogenous mixture, we use a magnetic stirrer while selecting samples. We compared measured GSDs from all five subsamples to ensure that each was uniform and representative of the entire population. We then merge the grain size data for the coarse grains from sieving with the fine grains from the particle analyzer by normalizing the volume fraction for each by the total volume lost during the experiment, calculated from measured mass and density values. Following the method used by the particle analyzer, the distribution is converted from volume fraction to number fraction by assuming

the grains are spheres.

## 3   Results

We conducted the binary collision experiments on a total of 5 sets of bricks, 2 sets of quartz diorite, and one set of each for the sandstone, schist, and volcaniclastic rock. Throughout the course of each experimental run, the initially cuboid rocks would quickly lose their sharp corners and then slowly become rounder without any major fragmentation. There were two exceptions

to this case. First, for the sandstone at around 20,000 collisions, a large piece roughly 2-cm long and 1-cm wide broke off one of the corners, exposing a reddish-orange oxidized surface. Second, with the schist on three occasions, the entire block more or less split in two, fracturing at weathering planes. In both cases, fracturing occurred at a pre-existing weak region of the rock that appeared to be associated with chemically-weathered surfaces. Furthermore, for both sandstone and schist we observed that immediately following the large fracture events, the mass loss of the parent grain would increase as the freshly exposed

rough surface of the grain smoothed.

We define cumulative mass loss as $M = \Sigma_{n=1...N}\Delta M$, where $N$ is the cumulative number of collisions, and equivalently cumulative impact energy as $E = \Sigma_{n=1...N}\Delta E$. Plots of cumulative mass loss against cumulative impact energy for all rock types show two distinct patterns: an initial rapid phase of mass loss that is similar for all lithologies and impact energies, followed by a transition to a slower, linear mass loss curve whose slope varies with rock type (Fig. 5a). To verify the functional

relationship between mass loss and energy while controlling for material properties, we performed experiments with three different masses of brick, spanning a range of collision energies of 0.04-0.22 J. Mass loss curves for all experiments are in good agreement with each other, and with a single linear trend (Fig. 6). Linear fits were then made to all mass loss curves,



resulting in the relation:

$$M/M_0 = kE + b. \tag{7}$$

To test the robustness of the linear fit, we generated a plot of $M/M_0 - b$ versus $kE$; where $b$ would be dimensionless and $k$ would have units of inverse energy. The quantity $kE$ is analogous to $E/E_s$, where $E_s$ is hypothesized to be a critical energy for chipping or fragmentation to occur. The collapse of data for all experiments shows that a linear relation is reasonable, but as anticipated fails to fit the initially-steep portion of the mass loss curve (Fig. 5b). We want to relate the two parameters in the linear fit (eq. 7) to physically-meaningful quantities. We turn first to the slope $k$, which controls the long-term attrition rate

for a given energy and should be controlled by material properties — and hence be related to $A$. Data indicate that the fitting parameter $k$ is directly proportional to the long-term attrition rate as described by the total mass loss divided by cumulative impact energy, $dM/dE$ (Fig. 7a), with slope of 1. This direct relationship indicates that the slope to the linear fit data, $k$, nicely characterizes the long term attrition rates for all of the lithologies explored in this study.

We can then examine the relationship between the Attrition Numbers and the long-term abrasion rates for each lithology.

The brittle Attrition Number $A_b$ is plotted against long term attrition rates $dM/dE$ (Fig. 7b) and demonstrate good correlation indicating that the brittle Attrition Number incorporates appropriate material properties to describe the long-term attrition of different lithologies. The semi-brittle Attrition Number $A_s$ varies widely and does not correlate strongly with observed abrasion rates $dM/dE$ (Fig. 7b-inset); we do not consider this parameter further in our analysis.

We now turn to the intercept. We find that the value $b$ in relation (7) is related to the quantity of pebble mass that is lost

before attrition reaches the slower, linear portion of the mass loss curve (Fig. 7c). In other words, it is the amount of attrition that occurs in the rapid, first portion. The parameter $b$ is related to the initial mass of each particle, with an average value of $b = M/M_0 = 0.0018$ and is approximately constant for all experiments (Fig. 7c). This result suggests that all particles transition to the slower, linear portion of the mass loss curve when they have lost a certain fraction of mass. Since collision energies and rock strengths are different, the only factor common to all experiments is particle shape; all particles were initially

cuboids. To test whether $b$ is related to shape, we plot the evolution of corner curvature and mass against cumulative energy (Fig. 7c); results show that the former tracks the latter, and becomes approximately constant when rock mass $M/M_0 = 0.0018$. This value is the same as $b$, meaning that curvature of corners becomes constant when the fraction of mass lost is equal to $b = M/M_0$.

By putting together the brittle Attrition Number and initial mass corresponding to $k$ and $b$, the attrition relation for mass loss

versus impact energy is:

$$M/M_0 = C_1 \frac{\rho Y}{\sigma^2} E + C_2 = C_1 A_b E + C_2 \tag{8}$$

where

$$C_1 = 7.1 \times 10^{-6} \quad C_2 = 0.0018 \tag{9}$$

For the case when $M \gg 0.0018 M_0$, the abrasion relation reduces to

$$M = C_1 A_b E \rightarrow \frac{M}{E} = C_1 A_b \tag{10}$$



This brittle attrition relation suggests that when the sharp edges are worn away, the attrition rate is directly proportional to the brittle attrition number multiplied by the constant $C_1$.

The SEM images show a considerable amount of damage in the region near the edge of the grains (Fig. 5b/c). This damage is characterized by large cracks that span parallel to the collision surface with smaller cracks branching perpendicular to them. In some instances, these cracks produced from impact intersect with inherent cracks or grain boundaries of the material, extending the damage zone further into the interior of the grain. The results of the damage zone length measurements are plotted in Fig. 8. Note that the measured distributions of crack lengths from the SEM images are unreliable in the small length limit due to image resolution. On the other hand, the large-length limit is an order of magnitude larger than the smallest resolvable length, so these measurements are dependable. Further, for the thin sections imaged, we were only examining a single 2-dimensional plane of a 3-dimensional object. Therefore, a measured crack length is a result of both the actual crack length and its orientation, as the length of the cracks running obliquely to the thin section plane will be underestimated. The tail of the distribution of lengths shows power-law scaling with exponents that range in value from -1 to -2.3. We observe convergence of all distributions for each lithology in the large length limit, where the cracks are easiest to discern and measure. However, in the lower length limit, the distributions diverge as the length measurements become less reliable due to the resolution of the images.

The results from the characterization of the GSD of attrition products are shown in Fig. 9. The plot combines the full measurements from the laser particle analyzer and sieving methods. Distributions from all lithologies and experimental runs show the same functional form. However, the full distributions display artifacts of the measuring techniques in both the fine and coarse tails of the distributions. For the fine tail, the distributions drop off rapidly, presumably due to the combined effects of the low-end measuring limit of the particle analyzer and the loss of material during the collection of attrition products. For the coarse end of the Coulter counter data, sieving produces artifacts in the grain size distributions as the particle size approaches the sieve diameter, as is evident by the erratic fluctuations in the grain size distributions on approach to $d = 0.5$ mm. Ignoring Coulter counter data over the range 0.2-1.0 mm, we observe consistent and smooth grain size distributions from 1 $\mu$m to the maximum observed size from sieve analysis, for all rock types. To determine the functional form of the grain size data, we remove the unreliable data points that are biased by the measurement method; for the fine tail, this includes grain sizes less than 1 $\mu$m, and for the coarse tail includes particle analyzer data greater than 200 $\mu$m. We normalize each curve by its mean value, collapsing all curves onto each other so that we may fit one function to the entire data set for all lithologies. We then solve for the best fit power law to all data points. The fit shows an exponent of 2.5, which is slightly higher than the expectation (2) for full fragmentation, but still follows a Weibull distribution with very good agreement (Fig. 9).

## 4 Discussion

While a linear relation between mass loss and impact energy has been shown to reasonably model aeolian erosion (Anderson, 1986), and has been inferred in models of bedrock erosion (Sklar and Dietrich, 1998, 2004), our experiments definitively demonstrate that this linear relation is applicable for energies associated with fluvial bed load transport, over a wide range of rock strengths. There is an intriguing shape dependence of the initial attrition rate. Indeed, data seem to indicate that these





initially very angular cubes all erode at the same rate regardless of energy or strength, until the corners are suitably rounded

such that energy and rock strength become important. We surmise that in this region the corners are so sharp that virtually any impact can remove mass, because the yield stress will be locally exceeded in the limit of infinite curvature (Ghadiri and Zhang, 2002a). However, rocks achieve the secondary linear mass loss curve quickly while their shapes are still very close to cuboids. Thus, for natural streams it is likely a reasonable assumption that $b$ may be neglected; therefore, the relation $M/(M_0E) = k = C_1A$ is the applicable one to examine attrition in natural streams.

The slope $k$ has units of 1/energy, and thus $1/k$ may be generically interpreted as a critical energy associated with breakage for each material. How energy relates to breakage depends on the failure mechanism; in particular, how elastic or plastic is the deformation associated with collision (Momber, 2004b). We examined two different formulations for the Attrition Number, $A$. It appears our data are reasonably well described by $A_b$ and not by $A_s$, indicating that material failure may be considered to be in the brittle regime. While previous work showed that bedrock erosion rate depends on the inverse square of the tensile

strength (Sklar and Dietrich, 2004), our experiments elucidate clearly and simply which rock material properties need to be taken into account through the development and verification of $A_b$. A similar attrition parameter was proposed by Wang et al. (2011) for the erosion of yardangs by windblown sand, but the material control on attrition rate was not isolated from collision energy in their work. Moreover, here we verify the concept for energies relevant to fluvial transport. Wang et al. (2011) noted that the parameter $A_b$ can be considered to be the elastic potential energy per unit volume at the yield point. We note, however,

the existence of the prefactor $C_1$, which at present is an empirical parameter derived from our particular experimental setup. The physical meaning of $C_1$ likely combines a few factors, certainly including the details of the collision itself; the impact angle, rotation speed of the impacter, and other aspects of the collision geometry (Wang et al., 2011). The value of $C_1$ may also be related to particle shape, although experiments by Domokos et al. (2014) show that $M/E$ is constant for a given particle over nearly the entire evolution from cuboid to sphere, suggesting perhaps that $C_1$ is independent of shape. Our data

tentatively suggest that $C_1$ is independent of material properties, since it is (roughly) constant across a range of material properties. Regardless, the brittle Attrition Number $A_b$ appears to be a useful similarity criterion for comparing laboratory and field attrition rates; however rates determined from our experiments may not yet be directly scalable to the field due to uncertainty in the controls on $C_1$.

While the brittle Attrition Number appears to describe the scaling of mass loss by chipping reasonably well, this provides

still an incomplete picture. In particular, the actual value of mass or volume removed per impact must be calibrated with experiment. The SEM images of sectioned rocks show a zone of damage accumulation in a shallow region below the surface. Our measurements show some isolated, surface-normal cracks that penetrate several hundreds of microns below the surface (Fig. 8). More common, however, are shattered regions of surface rock, that are bounded from below by surface-parallel cracks at depths of a few hundred microns (Figs. 4; 8d, 8e). This is important because lateral cracks are known to produce chipping

for natural rocks (Momber, 2004a). Our examination of the damaged rock took place after thousands of collisions, so we do not know what the damage zone from a single impact looks like. Our observations could be explained, however, by the merger of Hertzian fracture cones that often form in ceramics and glasses (Wilshaw, 1971; Greeley and Iversen, 1987; Rhee, 2001; Mohajerani and Spelt, 2010); it has also been claimed that such fracture cones explain the surface texture of sediment grains



(Greeley and Iversen, 1987; Johnson et al., 1989). In this model, elastic wave propagation outward from the impact site shatters
rock in a small region in which yield is exceeded (Wilshaw, 1971; Rhee, 2001). While still a brittle response, this failure mode
is distinct from the cyclic fatigue mechanism that has been proposed to activate slow growth of cracks in fluvial bedrock
erosion (Sklar and Dietrich, 1998, 2004). In high-resolution simulations of impacts on silica glass (Wang et al., 2017) — with
energies and material properties comparable to our quartz diorite collisions — formation of Hertzian cones produced locally
shattered near-surface regions that appear similar in style and scale to our images (Fig. 8). Importantly, this process is known
to generate small chips (Rhee, 2001; Mohajerani and Spelt, 2010; Wang et al., 2017). We tentatively conclude that chipping in
our experiments — and in pebble rounding and bedrock erosion, generally — is dominated by Hertzian fracture. We note that
Hertzian fracture has already been proposed as the primary mechanism for rock erosion by aeolian saltation impact, (Greeley
and Iversen, 1987), but was not examined in detail. Fatigue loading and associated slow fracture growth likely contribute to
fragmentation that produces larger particles. In our experiments such fragmentation was rare, but did occasionally occur; it is
likely to be more common in weaker rocks that are highly weathered (see below), or for higher collision energies.

Maximum measured crack lengths, and damage zone depths, are comparable to the maximum size of attrition products in
our experiments (Figs. 8, 9). Both crack-length and attrtition-product size distributions are power law, though scaling of the
former varies among materials and may not be reliable due to measurement limitations. We tentatively suggest that the attrition
product GSD is produced directly by localized (Hertzian) impact shattering, but acknowledge that more work is needed at the
individual collision scale — in particular, examining the shattered impact region of a rock in 3D. It is somewhat surprising
that maximum crack and attrition-product sizes vary little across all lithologies. Hertzian fracture cone size should depend
on material properties such as fracture toughness and Young's modulus, and also on the applied load (Wilshaw, 1971; Rhee,
2001; Mohajerani and Spelt, 2010; Wang et al., 2017). We speculate that more dynamic range is needed, in terms of both
material properties and impact energies, to see the effects of these factors. We also acknowledge our limited measurements,
due to the experimental challenges, which make our estimated maximum sizes unreliable. Nonetheless, the GSDs of attrition
products are well resolved, covering four orders of magnitude in size, and their similar form across lithologies demands a
generic explanation. A classic model for understanding GSDs resulting from wear is the brittle fracture theory developed by
Griffith (1921), who hypothesized that all materials contain pre-existing flaws or cracks. The theory states that when an applied
stress exceeds a critical value, the concentrated stress at the tips of these cracks is released as the crack propagates. Growth
and intersection of these cracks cause the ultimate failure of the material. In the large energy limit of crushing, where complete
disintegration of the parent particle occurs, Gilvarry and Bergstrom (1961) showed that the Griffith fracture model implies that
the attrition products should have a GSD that follows the form of eq. (5). More recent numerical simulations and laboratory
experiments have shown that the value of the exponent depends on the mechanism of fracture (i.e. grinding, collision, or
expansive explosion) and the impact energy (Kun and Herrmann, 1999; Astrom et al., 2004; Kok, 2011). However, none of
these studies examined the low-energy limit of chipping that is relevant for bed-load transport. The scaling exponent of 2.5
for the attrition products of these binary collision experiments is surprisingly robust across a range of rock types, indicating
a commonality in the failure modes of these different materials under the energies examined. The exponent is also within
the range of values reported from studies of brittle fracture fragmentation. These observations support the notion that brittle





fracture is the mechanism that creates the products of attrition in our experiments. The large-size limit seems governed by the
depth of the damage zone. As for the lower size limit, an obvious candidate would be the size of constituent particles in each
rock type; i.e., sand grains for the sandstone or clay particles for the brick. Although we could not resolve the finest particles
owing to loss, it is clear that fragmentation through constituent particles occurs. The determinant of the lower size limit remains
unknown. Nonetheless, chipping robustly produces sand and silt sized particles, supporting the proposal that it is an important
contributor of sediment to rivers and beaches (Jerolmack and Brzinski, 2010).

In the limit where $k = 0$, the brittle Attrition Number, $A_b$, does not likewise approach zero, but instead is associated with
$A_b = 0.25$. This non-vanishing value of $A_b$ implies that for the range of energies examined in this experiment, there is a
limiting rock strength at which little or no attrition occurs. (This is similar to the proposed lower limit for collision energy,
below which chipping does not occur (Ghadiri and Zhang, 2002a; Novak-Szabo et al., 2018).) This result would suggest that
some materials should not erode significantly under impact energies representative of bed load transport. For our experiments,
the volcaniclastic rocks are close to this limit. Observations of downstream evolution of pebble shape for volcaniclastic rocks
in the River Mameyes in Puerto Rico have shown that significant attrition occurs (Litwin Miller et al., 2014). However, the
pebbles from the field were all at least 4 times larger than those used in the laboratory, while estimated collision velocities were
comparable. The combined observations of volcaniclastic rocks from experiments and the field suggest the possibility that, as
particles lose mass downstream due to chipping, there is a potential lower limit in size that is controlled by rock strength. This
idea needs to be explored in more detail.

Although results from these experiments display a steady linear mass loss with impact energy, as evident with the large
fracture events with the sandstone and schist, chemical weathering can play an important role in the breakdown of river
sediment. Howard (1998) observed higher rates of bedrock erosion in regions with more chemical weathering and thereby
showed that chemical weathering weakens rocks and reduces material strength. While we find that material properties control
attrition rates, chemical weathering can cause a weakening of these material properties. We observe fragmentation events along
weathering planes, similar to those observed in experiments of Kodama (1994b). In these instances, new angular and rough
surfaces produced from the fragmentation process have high attrition rates. On the one hand, chemical weathering appeared
to create internal planes of weakness that facilitated failure of large chunks under low-energy attrition. Indeed, these events
caused fluctuations in the mass loss curves that were not observed in more structurally sound (stronger) materials. However,
when observed over thousands of collisions (i.e., many fracture/failure events), the sandstone and schist rocks collapsed onto
the same linear curve as other lithologies after accounting for material strength. It appears that mechanical weakening from
chemical weathering may be reasonably described with the measured material properties that constitute $A_b$, so long as tested
rock cores are representative of the rocks in question. In a natural setting, we expect that the effects of chemical weathering
will be more dominant in transport-limited environments where chemical weathering rates outpace mechanical attrition. On
the contrary, where sediment is transported frequently, mechanical wear is actively maintaining fresh unweathered surfaces on
rocks, and therefore weathering features are not able to persist. Finally, we note that impact attrition experiments show that
water may reduce the strength of silicate materials by half compared to measurements in air (Johnson et al., 1973). This is





expected to influence the rate but not the style of attrition, and serves as another caveat in applying our measured results to the field.

## 5 Conclusions

The results of this laboratory investigation suggest that the main consequences of fluvial attrition are encapsulated in two "universal" relations. First, we verified a linear mass-loss relation for energies and particle sizes associated with fluvial transport. In doing so, we have shown which material properties control the amount of mass loss per unit energy, providing a mechanistic underpinning to attrition "Susceptibility" (Anderson, 1986) and helping guide researchers regarding how to characterize lithology's control on attrition. Second, the grain size distributions for attrition products suggest that brittle fracture creates fragmentation over a restricted skin depth, that may be associated with Hertzian fracture cones. More theoretical and single-impact experimental work is necessary, however, to understand the underlying mechanics of fracture and damage. More pointedly, measurements of fracture toughness for rock surfaces due to impact — rather than the usual estimates from bulk fracture under static loads — are needed; Hertzian fracture tests could be a useful technique for this (Wilshaw, 1971). In addition, we have identified a possible shape control on attrition rate in the initial stage where particles are very angular. This is intriguing from a mechanics point of view, but is likely negligible in nature as the effect is only manifest when corners are exceedingly sharp.

Our experiments have shown that material properties can be accounted for reasonably simply; however, results cannot be scaled directly to the field until the constant $C_1$ is understood. We hypothesize that this coefficient is primarily controlled by the details of the collision process, which determine how much impact energy contributes to damage as opposed to friction or rebound of the target. Once $C_1$ is resolved, one may use a mechanistic model of bed-load collision energy and frequency to estimate attrition rates in natural rivers. If the grain size distributions of attrition products are indeed universal, they could also be used to estimate the quantities of sand, silt and dust that result from attrition by bed load transport. If the results of Domokos et al. (2014) and Litwin Miller et al. (2014) are correct that up to 50% of a pebble's mass is lost during transport downstream, significant quantities of these fine grains are produced in natural rivers.

*Data availability.* All data used for plots in this paper are deposited on figshare, a publicly available and open repository.

*Author contributions.* K.L.M led the research, performed data analysis and wrote the paper. D.J.J. supervised the research, assisted in data interpretation, and edited the paper.

*Competing interests.* The authors declare no competing interest.





*Acknowledgements.* We thank G. Salter and R. Martin for their assistance in the laboratory. Thank you to L. Sklar for allowing use of

450    equipment and lab space for material property measurements and Sirui Ma for hardness measurements. We thank the editor and reviewers

of the manuscript for their time and feedback. We acknowledge the Luquillo Critical Zone Observatory (NSF EAR 1331841) for financial

support.





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



Earth **Surface**
**Dynamics**
Discussions

EGU

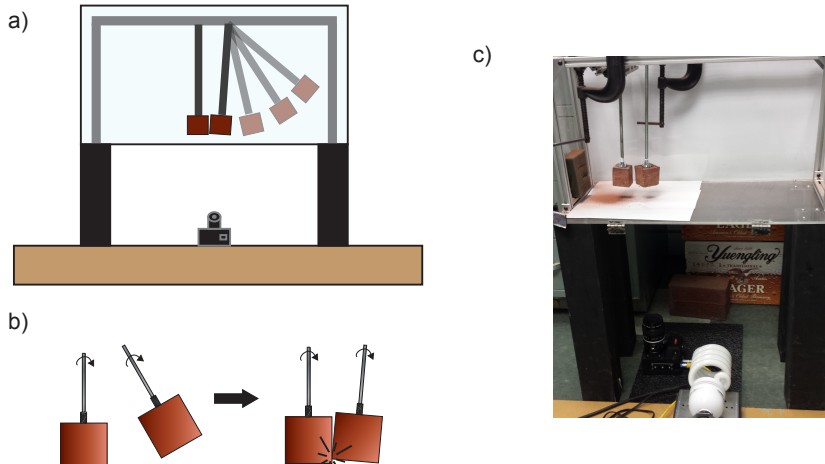

**Figure 1.** Schematic of experimental set-up. (a) Front view drawing depicting binary collisions double-pendulum apparatus. (b) Close-up drawing illustrating how grains impact during collision. The impacting grain is raised then released, colliding with the stationary target grain. Both grains are able to rotate freely. (c) Picture of set-up with brick clasts.





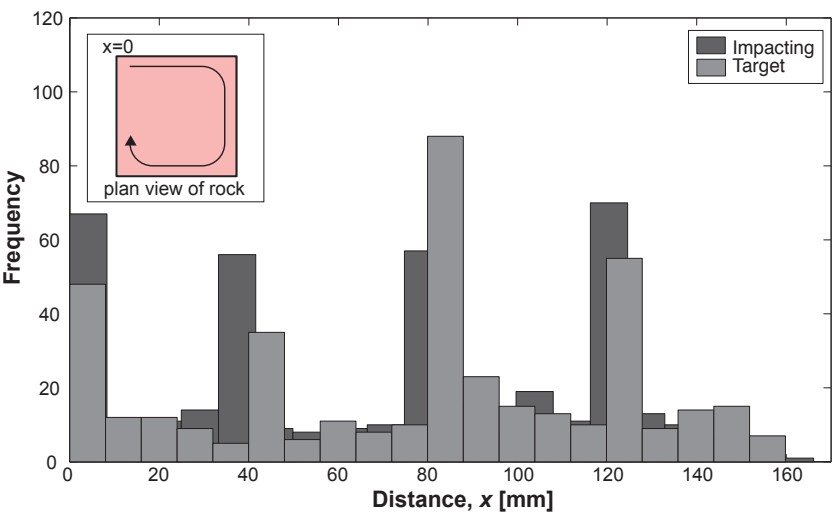

**Figure 2.** Randomness test for collision rotation. Plot showing the histogram of impact locations for impacting and target grains. Peaks correspond to corners of the grains. Inset shows plane view of rock with labeled location of $x = 0$ at one of the corners.





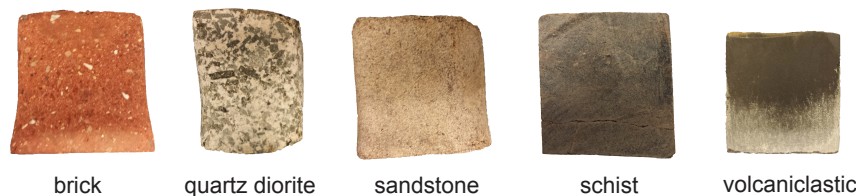

brick    quartz diorite    sandstone    schist    volcaniclastic

**Figure 3.** Images of samples. Images of all rocks used in the experiments. Images taken at the end of each experiment so there is noticeable rounding of the edges.





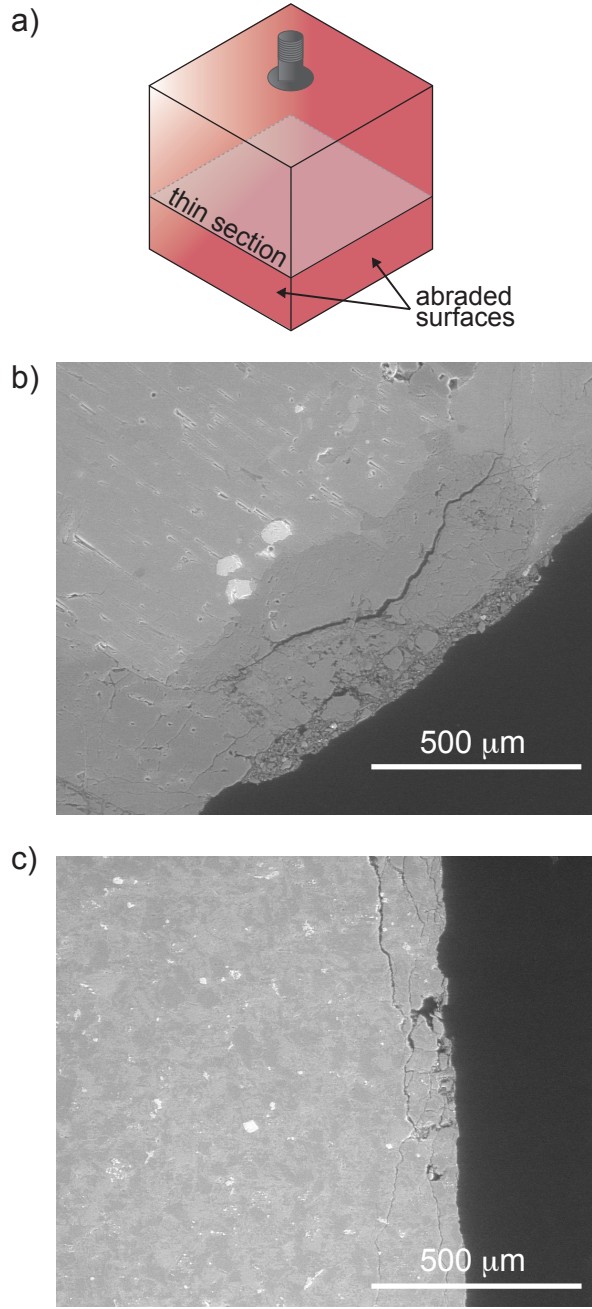

**Figure 4.** Thin section preparation and SEM images. (a) Schematic drawing showing location in grain where thin sections were made. (b) SEM image of quartz diorite. (c) SEM image of volcaniclastic rock.





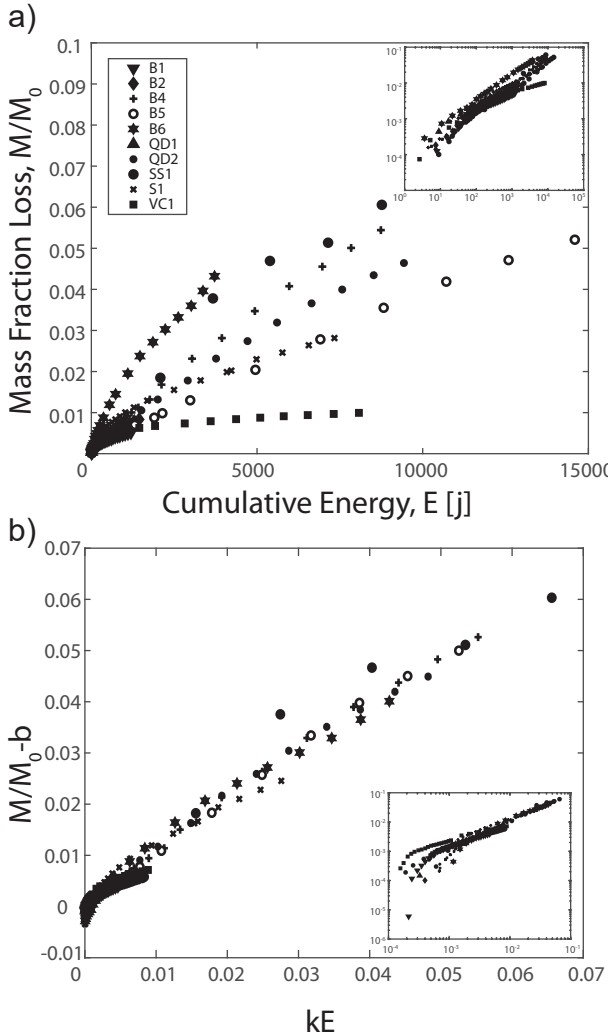

**Figure 5.** Abrasion mass loss curves. (a) Plot of total cumulative mass loss versus cumulative impact energy for each set of rocks. (b) Plot of total cumulative mass loss minus y-intercept, $b$, from linear fits to raw data in (a) versus cumulative impact energy multiplied by value of fit slope. Insets for both (a) and (b) displays plots with log-log axes.



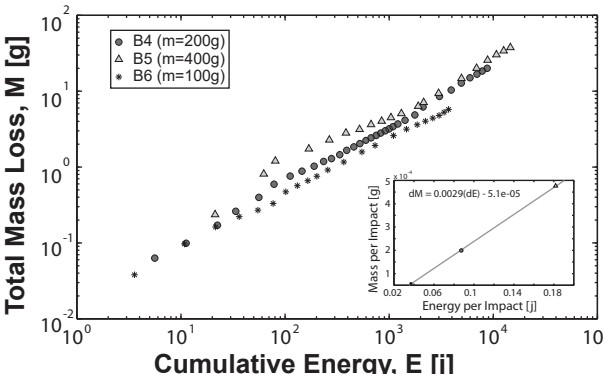

**Figure 6.** Abrasion rate for bricks with different collision energies. Plot of total cumulative mass abraded versus cumulative impact energy for three sets of brick with different masses. Inset displays plot of average mass abraded per impact versus average energy per impact. Each data point corresponds to a separate set of bricks.

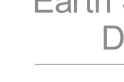
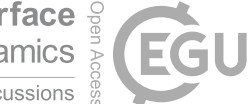


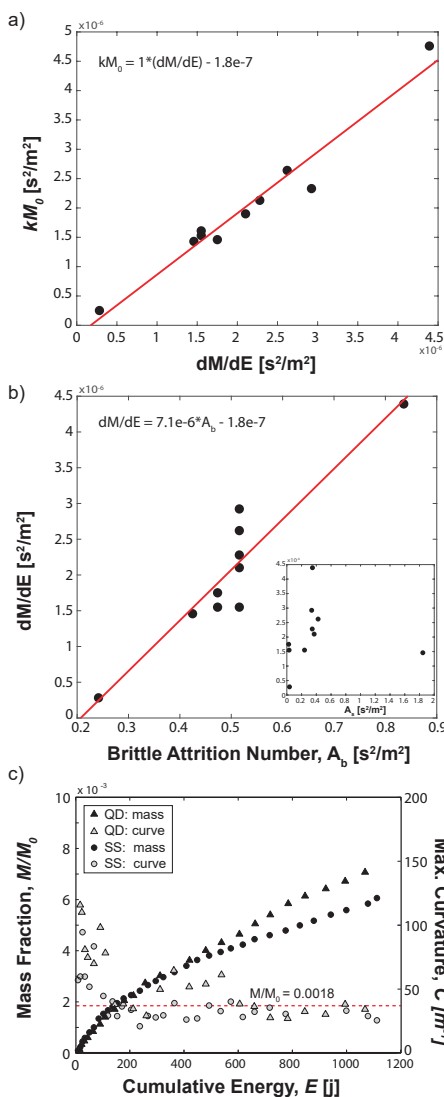

**Figure 7.** Attrition Numbers. (a) Plot of slope($k$) from linear fits to normalized mass curves multiplied by initial mass versus total mass loss over total energy. (b) Plot of dM/dE versus brittle attrition number. Inset is plot of calculated semi-brittle attrition number. (c) Plot showing change in mass fraction (left axis) and maximum curvature(right axis) versus cumulative impact energy. They both transition from a high rate of change to a slower one at average intercept value of $M/M_0 = 0.0018$.





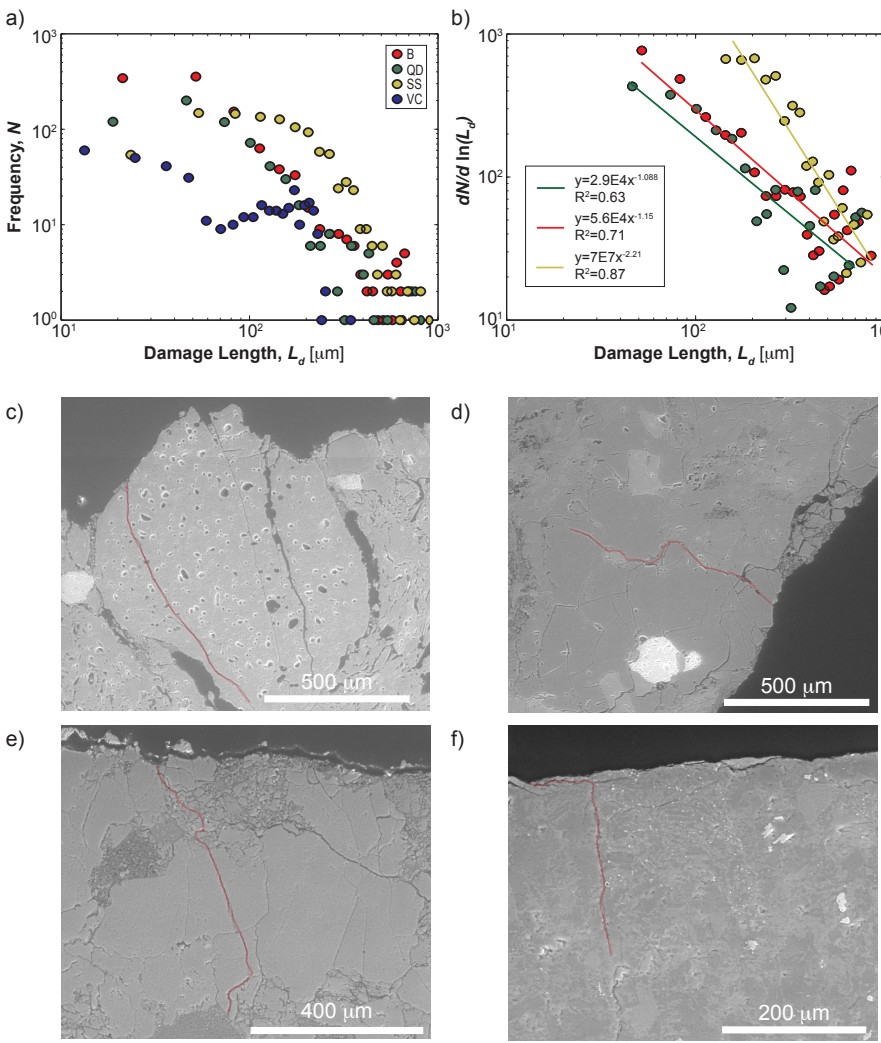

**Figure 8.** SEM results. (a) Plot of the distribution of length of damage within abraded rocks from SEM images of thin sections. (b) Damage lengths plotted in the form of the eq. (5) with corresponding power-law fits. (c)-(f) SEM images of the largest crack length for each rock type outlined in red. (c) Brick (d) Quartz diorite (e) Sandstone (f) Volcaniclastic rock



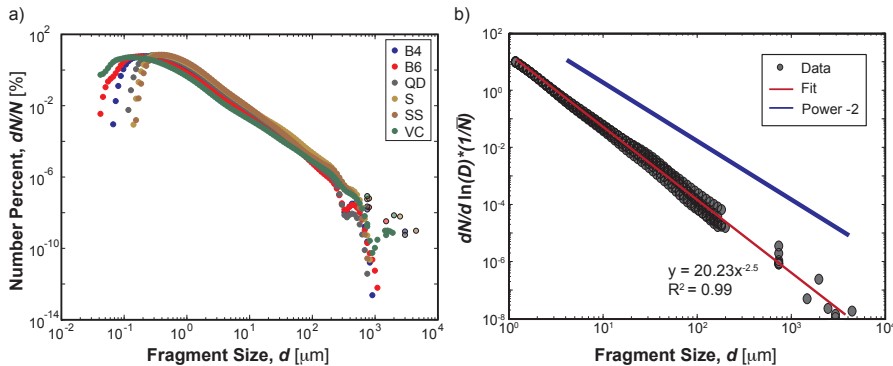

**Figure 9.** Grain size distributions of products of attrition. (a) Plot of number distribution of grain size from particle analyzer (solid circles) and sieving (circles with black outlines) methods. (b) Number distribution of grain size plotted in the form of eq. (3) and normalized by the mean value. Data are combined for all lithologies. Data believed to be affected by measuring technique are excluded from plot ($< 1 \ \mu$m and $> 200 \ \mu$m from particle analyzer). Data fit with power law function with exponent of $-2.5$. Solid blue line denotes expectation from brittle fragmentation, power law with exponent -2.



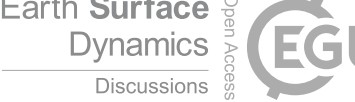

| Sample | Lithology | Initial Size $D$ [m] | Density $\rho$ [kg/m$^3$] | Tensile Str. $\sigma_t$ [MPa] | Youngs Mod. $Y$ [GPa] | Int. Mass $M_0$ [g] | Impact Energy $E_i$ [j] | Slope $k$ [j$^{-1}$] |
|---|---|---|---|---|---|---|---|---|
| B1 | brick | 0.0618 | 2072 | 7.5 | 14 | 490 | 0.159 | 3.88E-6 |
| B2 | brick | 0.0575 | 2072 | 7.5 | 14 | 394 | 0.148 | 5.40E-6 |
| B4 | brick | 0.0562 | 2072 | 7.5 | 14 | 369 | 0.112 | 6.32E-6 |
| B5 | brick | 0.0707 | 2072 | 7.5 | 14 | 734 | 0.213 | 3.60E-6 |
| B6 | brick | 0.0400 | 2072 | 7.5 | 14 | 133 | 0.036 | 1.15E-5 |
| QD1 | quartz diortie | 0.0395 | 2704 | 16.9 | 20-70 | 167 | 0.090 | 8.76E-6 |
| QD2 | quartz diorite | 0.0488 | 2704 | 16.9 | 20-70 | 315 | 0.148 | 5.11E-6 |
| SS1 | sandstone | 0.0649 | 2330 | 5.28 | 10 | 636 | 0.179 | 7.48E-6 |
| S1 | schist | 0.0522 | 2667 | 20.5 | 7 | 381 | 0.091 | 3.76E-6 |
| VC1 | volcaniclastic | 0.0441 | 2672 | 6.63 | 5-50 | 229 | 0.052 | 1.10E-6 |

**Table 1.** Table listing measured material properties and experimental conditions for each set of samples