# Peer review of "Controls on the rates and products of particle attrition by bed-load collisions"

_Earth Surface Dynamics, 2020_

## Short Comment (SC1) · 9 Dec 2020

**Notes on Miller-Jerolmack Particle Attrition**

This is not a formal review of this paper. Rather, please view the comments below as notes to myself, aimed at my trying to understand elements of the analysis.

The experiments and analyses reported in this paper are clever.

1) Because in physics and mathematics a "double pendulum" has very specific meaning and significance,

https://en.wikipedia.org/wiki/Double_pendulum ,

the authors might consider using "dual pendulum" or "paired pendulum."

2) The presentation by Kok (2011) is a bit misleading. A value of -2 is not claimed by Gilvarry and Bergstrom (1961), and Astrom (2006) actually focuses on fracture size, not fragment size. Eq. (1) in Kok (2011) is a power-law distribution. His Eq. (2) is sort of like a Weibull distribution, but is not. Moreover, it is incorrect for Kok (2011) to say that $N_f$ represents a number of fragments. It is in fact a number density.

Here is my take on where Eq. (2) in Kok (2011) comes from. To simplify notation, let $N_f(D_f) \to n_x(x)$ with $D_f \to x$. Now let $f_x(x)$ denote the probability density function of the fragment sizes $x$, and let $N$ denote the total number of fragments. Then

$$n_x(x) = N f_x(x) \tag{1}$$

is the number density of fragment sizes. Now

$$\mathrm{d}n_x(x) = N f_x(x)\,\mathrm{d}x \tag{2}$$

is interpreted as the number of fragments within the small interval $x$ to $x + \mathrm{d}x$. (Note that this wording is consistent with that of Gilvarry (1961).)

We now write

$$y = \ln x \tag{3}$$

In turn,

$$\mathrm{d}y = \mathrm{d}(\ln x) = \frac{\mathrm{d}x}{x} \tag{4}$$

Note that the parentheses are implied in writing $\mathrm{d}\ln x = \mathrm{d}(\ln x)$, as in Kok (2011). This means that $\mathrm{d}x = x\,\mathrm{d}\ln x$. Whereas $\mathrm{d}x$ denotes a small increment of $x$ in arithmetic space, $\mathrm{d}\ln x$ denotes a small increment of log space. Now,

$$\mathrm{d}n_x(x) = N x f_x(x)\,\mathrm{d}\ln x \tag{5}$$

That is,

$$\frac{\mathrm{d}n_x(x)}{\mathrm{d}\ln x} = N x f_x(x) \tag{6}$$

The power law distribution in Eq. (1) of Kok (2011) in effect neglects the exponential part of his Eq. (2). Moreover, if his Eq. (1) is correct, then it implies that $f_x(x) \sim x^{-3}$ such that the cumulative distribution goes as $x^{-2}$. This is a rather heavy-tailed distribution whose mean may be undefined. Gilvarry and Bergstrom (1961) find that the cumulative distribution of fragment volume $v$ (rather than linear size $x$) goes as $v^{-2/3}$.

By writing the distribution as in Eq. (6) above (like Eq. (2) in Kok (2011)), the data can be fit using bins of specified log increment — as with a histogram — rather than the usual manner of constructing an exceedance probability plot. This is rather clever, as it avoids the issue of censorship, which, if not addressed in constructing empirical exceedance probabilities, can change the slope of the fitted line. The caveat is that $N$ must be relatively large to support the binning.

That said, I hope I am not misinterpreting how Miller and Jerolmack constructed the plot in Figure 9.

djf

---

## Referee Comment (RC1) · Stephanie Deboeuf (Referee) · 7 Jan 2021

The work of Miller and Jerolmack entitled 'Controls on the rates and products of particle attrition by bed-load collisions' deals with earth surface dynamics, by taking into account fracture mechanics, leading to a very interesting and well treated multidisciplinary approach.

The authors realized well controled experiments of particle impact and attrition and clever data analysis, as well as precise size measurements of attrition products, allowing them to get their main experimental results: impact erosion can be treated 'as brittle fracture in the purely elastic regime'. Additionally, their fine observations of chipped

particles allow them to support that 'the common fatigue failure model is inappropriate', but 'propose that Hertzian fracture is the dominant mechanism'. Again, materials mechanics appear surprisingly as a relevant tool for bedrock erosion, sand production, bed-load transport, ... The authors also consider the limitations of the methods and take time to explain them to the readers, that is really appreciated.

The whole work is realized rigorously. High numbers of different experiments are done to ensure good statistics (450 collisions to test the randomness of the grain rotation, 50 to 10 000 collisions, 20 000 collisions, ...), that is really appreciated. I also really appreciate the analysis of experimental data by using dimensional analysis and knowledge from elasto-plasticity, as well as the desire of the authors to use 'physically-meaningful quantities'.

For all these reasons, I agree the publication of the paper. However, I have some suggestions reported in sequence in the pdf file.

Please also note the supplement to this comment:
https://esurf.copernicus.org/preprints/esurf-2020-86/esurf-2020-86-RC1-supplement.pdf

**Supplement:**

The work of Miller and Jerolmack entitled « Controls on the rates and products of particle attrition by bed-load collisions » deals with **earth surface dynamics**, by taking into account **fracture mechanics**, leading to a very interesting and well treated **multi-disciplinary approach**.

The authors realized **well controled experiments** of **particle impact and attrition** and **clever data analysis**, as well as precise size measurements of attrition products, allowing them to get their main experimental results: **impact erosion can be treated « as brittle fracture in the purely elastic regime »**. Additionally, their **fine observations** of chipped particles allow them to support that « the common fatigue failure model is inappropriate », but « propose that Hertzian fracture is the dominant mechanism ». Again, **materials mechanics** appear surprisingly as a relevant tool for bedrock erosion, sand production, bed-load transport, … The authors also consider the limitations of the methods and take time to explain them to the readers, that is really appreciated.

The whole work is realized rigorously. High numbers of different experiments are done to ensure good statistics (450 collisions to test the randomness of the grain rotation, 50 to 10 000 collisions, 20 000 collisions, …), that is really appreciated. I also really appreciate the analysis of experimental data by using dimensional analysis and knowledge from elasto-plasticity, as well as the desire of the authors to use «  physically-meaningful quantities ».

For all these reasons, I agree the publication of the paper. However, I have some suggestions below that are reported in sequence.

Maybe it would be interesting to say a word about the case of an **impact of a 'pebble' with a granular material** (with or without cohesion), the last one as a model of some river beds. Such non consolidated (discrete) materials would be another class of materials, where there is not any true fracture, but where mass loss can occur. I know the papers from Beladjine et al 2007 (PRE), where they found a relation between the mass loss Nej, the effective restitution coefficient e function of the impact angle, and the Froude number. Are the experimental data or the current knowledge enough conclusive to be compared to your Eq (2) page 2: Delta M = A Delta E? However, maybe these experiments are not realized «  under conditions relevant for bed-load transport »?

I would like to see **some references about papers from the mechanical community about brittle fracture in the purely elastic regime**, if relevant for your work.

Main text

1) Introduction.

When I read **« attrition rate »** for the first time in your paper (line 52, page 2) and later (line 259 page 9), I am wondering to **which definition** you refer to: a time derivative of mass or a derivative of the mass according to the impact energy? I guess it is the derivative of the mass according to the impact energy, however it is not obvious in general.

**Maybe you should refer to the Charpy impact test, that look like your experimental set-up, even if boundary conditions are not the same.** With such a test, do you think that the measurement of absorbed and/or released energy during the impacts and rebounds would be possible and interesting for your open questions? I think this may be related to one of your

conclusion l. 440 p. 14 « We hypothesize that this coefficient is primarily controlled by the details of the collision process, which determine how much impact energy contributes to damage as opposed to friction or rebound of the target. »

2) Methods.

2-1)

Dimensional analysis is really appreciable, I think that it may be very useful.

However, whereas it is clear for the brittle/purely elastic regime, I find it a bit less clearly written for the semi-brittle case. I do not know if you should present the ratio $H/K_c$ as the third dimensionless group and/or you should explain why D, the sample diameter appears as an input parameter in the semi-brittle regime but not in the brittle regime (localized plastic deformations at the surface have to extend to the bulk, on a size D?)?
At this end, it is not crucial, because your experiments are well described by $A_b$, and not by $A_s$. But, this lacks.
Also, it is not clear **what are the units of parameters** appearing in $A_s$: what is the unit of H (and $K_c$), so that $A_s$ have the same units as $\Delta M / \Delta E$ and $A_b$? Nowhere in the paper, values of H are given, whereas $A_s$ is computed?

2-2)

How do we know typical impact velocities or energies involved in bed-load transport to state that the values reported here in the experiments are **« under conditions relevant for bed-load transport »****?**

l.176: Why the area coming into play in the expression of the tensile strength is pi l D /2 instead of pi D^2 / 4 ?

l. 208: The number $A_s$ is called here the Attrition Number for the first time, you should have introduce this name for $A_s$ and $A_b$, when these latter are introduced page 4.

3) Results

l. 252: In the sentence « Mass loss curves for all experiments are in good agreement with each other, and with a single linear trend (Fig. 6). », the group « a single linear trend » would suggest that the slope is the same, that is not true. Maybe you should add « with different slopes » or something like that.

I am wondering why you choose to write your Eq (7) page 9 as $M/M_0 = k E + b$ , instead of writing k as $1/E_s$; this would be more direct and this would avoid introducing two variables (k and $E_s$) instead of only one.

l. 260, I would make a remind here on what is A: the Attrition number.

l. 265: « The brittle Attrition Number $A_b$ is plotted against long term attrition rates dM/dE (Fig. 7b) and demonstrate good correlation », it should be added here that **there is some scattering**. Fig 7b: To which materials correspond the plotted data? Maybe, it should be written somewhere that $A_b$ is between 0 and 1 and $A_s$ between 0 and 2, thus are of the same order of magnitude. Say something on the precision or uncertainty of $A_b$ (and $A_s$). Say something about the slope of the order of $10^{-5}$. As it is very far from 1, it should mean that some physical understanding still lacks, that will need further investigations in the future.

l. 272: « The parameter b is related to the initial mass of each particle, with an average value of b =$M/M_0$ = 0.0018 and is approximately constant for all experiments (Fig. 7c). » **I don't see that b is approximately constant for all experiments in Fig. 7c**.

l. 276: « results show that the former tracks the latter, and becomes approximately constant when rock mass $M/M_0$ = 0.0018 » I would write instead « $M/M_0 \geq 0.0018$ »

l.277: « This value is the same as b , » **The value of b is not shown (in a Figure) or given (in Table 1). This lacks.**

l. 281: Why introducing a new symbol C2, since it is b? I would change the sense of presentation of Eq.(8) as:
$M/M0 = C1\ Ab\ E + C2 = C1\ rho\ Y\ E / sigma^2 + C2$
Also, it seems to me that there is a slight approximation because k is not equal to C1 Ab but is equal to C1 (Ab - 0.2) as can be seen in Fig. 7b. So, the term - 0.2 C1 E lacks in Eq(8). You should say a word about this.

l. 285: In accordance with Eqs (7) and (8), M should be divided by M0 in Eq (10). As a consequence, the sentence in l. 286 should be added by « and divided by M0 ».

l. 299: Instead of « diverge », I would use « differ » to avoid suggesting wrongly tend towards infinity.

l. 312: Add the reference to Eq (6) in the sentence « We then solve for the best fit power law to all data points. »

4) Discussion

l. 322: I would add that the sentence « However, rocks achieve the secondary linear mass loss curve quickly while their shapes are still very close to cuboids. concern rocks observed here » or something like that.

l. 324: I would begin a new paragraph to highlight the equation $M/(M0\ E) = k = C1\ A$ and I would change the first symbol = by the symbol $\simeq$.

l. 329: « It appears our data are reasonably well described by $A_b$ and not by $A_s$ , indicating that material failure may be considered **to be in the brittle regime. » This interesting conclusion stem from the correlation of your measurements with the brittle Attrition number AB: could we have infer it without your measurements, but from the values of elasto-properties (or from the comparison of the values of Ab and As)? If not, which data (impact stress?, impact deformation? …) should we have access to so that it becomes possible?**

l. 331: When using Ab here, I would recall its relation with Ab=rho Y / sigma^2.

l. 346: I would refer to Fig. 4 and 8 in the sentence « The SEM images of sectioned rocks show a zone of damage accumulation in a shallow region below the surface. »

l. 367: attrtition-product -> attrition-product

l. 395: « In the limit where k = 0 , the brittle Attrition Number, $A_b$ , does not likewise approach zero, but instead is associated with $A_b = 0.25$ . » It is not clear to me where this rationale comes from? **From Fig 7b, I can read that Ab(dM/dE=0)=0.25. But this not the value your reported.** Explain better, please. And what is the unit of Ab? Here it appears that it is without unit. But in Eq (4), Ab has the unit of time square / meter square. Clarify it please.

l. 402: « However, the pebbles from the field were all at least 4 times larger than those used in the laboratory, while estimated collision velocities were comparable. » Does it mean that your experiments are done at the same velocities as in the field, but at smaller energy? Do you think that normalizing the mass loss by the initial mass allows suggesting that your results to become independent on M0? Can you still write that you are in the same conditions as in the field (« under conditions relevant for bed-load transport »)?

Figures

Figure 1: I found it not clear at the beginning that grains were free to rotate, maybe your sketch Figure 1 should be completed with additional arrows to indicate the movements direction, and maybe some words should be added in the main text.

Figure 2: I would add x=40mm, 80mm, 120mm or add the word « square » to « rock » and one additional value x, so that it is clear that the distances x=0, 40, 80, 120mm correspond to the square rock corners.

Figure 5: Both lin-lin and log-log plots are really appreciated. You should cite your Eq (7) p. 9 in your legend, when talking about « linear fits ».

Figure 6: In your inset, you show average mass, we would like to see also some error bars describing the standard deviations of mass data. How your data shown in inset are compared to your Eq (7)?

Figure 7a: In the legend, you should add the precision « for all samples ».

Figure 7b: To which materials correspond the plotted data? Maybe, it should be written somewhere in the text that Ab is between 0 and 1 and As between 0 and 2, thus are of the same order of magnitude. Say something on the precision or uncertainty of Ab (and As).
Also, there is some scattering in the data: according to you, what is its origin?

Figure 7c: Write in your legend that data are for two samples: which ones?

Table 1: You shout add some properties here: Attrition numbers Ab and As, H and Kc? The tensile strength is referred to as sigma_t here, whereas in the whole text, one finds sigma, sigma_t, sigma_f, … one may get lost. Can you harmonize it please?

Sometimes, figures or equations are mis-referred:
p. 10 l. 288: Fig 5 b/c -> Fig 4 b/c
Figure 8: In the legend, Eq (5) -> Eq (6).
Figure 9: In the legend, Eq (3) -> Eq (6).

---

## Referee Comment (RC2) · Sebastien Carretier (Referee) · 19 Feb 2021

The authors present a very interesting contribution on pebble attrition. Based on a large number of pebble impact experiments, the authors conclude that size reduction rate is proportional to energy and occurs mainly during elastic impacts rather than by damage, with the production of a fairly fine material whose grain size is distributed as a heavy-tailed power law. This paper is very well written and illustrated and the conclusions are supported by robust statistics over a large number of samples. In particular, Figure 7b illustrates the main conclusion. I have only minor remarks.

The first concerns the relationship between the experimental set-up and the transport

of pebbles in a river. From what I understand (but I may be wrong) the shock is frontal in the experiments even if there is a certain degree of freedom in the pendulum movement. In rivers the pebbles can roll and rotate, with a tangential component in the shock. So I wonder if this difference may affect the transposition of the experimental results to natural rivers.

Looking at the pictures in thin sections, it seems that a damage zone forms well with a width that seems to be greater than the (potentially future) "grains" delimited by the fractures. This suggests that the formation of this band still partly controls the detachment of grains, and thus that damage plays a role. I recognize the hypothesis test on the basis of the Ab (brittle) and As (damage) parameters in Figure 7b, but I have difficulty in linking a purely brittle behaviour to this band which could be related to damage, although the authors recognize that more work is needed to verify that this band corresponds to a Hertzian fracture band.

Line 58 should be (e.g., Shipway and Hutchings, 1993).

Line 92. If deformation is "purely elastic" it would be reversible and thus without perennial cracks or fractures.

Line 355-358 A little more explanation of this difference here might help to better understand.

Line 408 See also Jones and Humphrey, 1997.

Best wishes, Sebastien Carretier

---

## Referee Comment (RC3) · Jeffrey P Prancevic (Referee) · 13 Mar 2021

Review of "Controls on the rates and products of particle attrition by bed-load collisions" by Kimberly Litwin Miller and Douglas Jerolmack

This manuscript presents the results of a fascinating set of experiments, and provides semi-empirical predictions for both rates of sediment attrition and the size distributions of the attrition products. The experiments were nicely designed and the measurements were detailed and complete (except in cases where certain material properties weren't possible to measure). The authors used a pair of pendulums to repeatedly collide two

grains of the same size and rock type. Characteristic velocities (and kinetic energies) were measured with high-speed cameras, and masses were measured between sets of collisions. SEM imagery of thin sections of particles after the experiments were used to observe and measure the development of fractures. Material properties were measured with various standard methods.

The manuscript presents several interesting findings, but, from my perspective, the most important claim is that the rate of mass loss (as a function of impact energies) can be predicted from material properties and empirical constants measured in this study. However, this claim is not adequately demonstrated for several reasons that are outlined below. These issues should be addressed before the manuscript is accepted for publication. Otherwise, the manuscript is well-written, easy to follow, and full of cool observations. I do list several minor points by line number below.

Issues with the prediction of the rate of mass loss

Equation (8) presents an elegant model of mass loss as a function of impact energy, material properties, and two empirical constants—C1 and C2. However, the observational basis for both C1 and C2 is shaky, based on the information presented in the manuscript.

C1 is based on the best-fit curve between Ab (based on material properties) and the ratio of mass lost to cumulative energy, shown in Figure 7b. This plot shows a cloud of 8 data points in the middle with no strong relationship, and two outliers–one with high Ab and one with small Ab. There is no legend to identify these data points, but based on information in Table 1, the small-Ab data point represents the volcanoclastic cubes. The authors were unfortunately not able to measure the Young's modulus for the volcanoclastic samples because they specimens were too short, and they instead rely on values from the literature, which span an order of magnitude: Y is between 5 and 50 GPa. Therefore, possible values for the Brittle Attrition Number (Ab, the x-axis of Figure 7b) also span an order of magnitude. Strangely, based on the Figure 7b and the

other material properties for the volcanoclastic rocks presented in Table 1, the authors used a value for Young's modulus that is outside the range found in the literature: Y = 4 GPa. The authors should double-check the values used in their calculations and be explicit about the material properties used to estimate Ab. For experiments where material properties are looked up from the literature, the manuscript should present a range of values of C1.

The issue with C2 is less important, particularly since the authors note that this early mass loss is mostly due to the cuboid shape of the particles and is likely unimportant for natural particles. Still, I was confused as to why Figure 7c only shows experimental results from the sandstone and quartz diorite experiments, while the manuscript claims that the early mass-loss behavior is universal. If it's truly universal, it would be more compelling to show that behavior for all of the particles.

Finally, Equation 8 is calculated from several steps, not one regression, and the predictive ability of Equation 8 is not tested. This model should be compared against the data shown in Figure 5a to show how well it predicts mass loss. This isn't a true test of the model, since it's comparing it against the data used to create the model, but it's better than nothing.

Comments by line number

23. Consider replacing "it" with "attrition" to avoid confusion with "abrasion"

58-73. I'm not an expert in fracture mechanics, so much of this discussion here and in the discussion section (354-365) was difficult for me to follow. That said, my reading of this section is that it is commonly assumed that attrition processes occur by the progressive development of cracks through the entire particle, rather than local fractures (surface parallel or otherwise). This certainly isn't the conceptual model that I would normally assume (I would local fracture around the impact site to be important), but if that's what people do normally assume then it's very good to point that out. Right now this statement of what is "typically assumed" is supported by only one reference from

60 year ago.

114-115. These sentences require the reader to differentiate between "significantly smaller" and "much smaller." Consider rewording.

262. Why are dM and dE used for cumulative values of mass loss and energy expenditure? Aren't M and E defined the same way? If there is a need to differentiate these values from M and E, consider using big delta, rather than the derivative.

313. Exponents should be negative (-2.5 and -2)

367. "attrition" spelling

Table 1. Consider adding columns for total number of impacts, Ab, and As.

Figure 7c. Show data from all experiments here

Figure 8b. Why are lengths not shown for the volcanoclastic rocks?

---

## Author Comment (AC1) · 20 May 2021

Dear Editorial Team,

We would like to thank you and the three reviewers for taking the time to provide comments on our manuscript. We have adopted nearly every suggestion made to improve the paper. We are very proud of the work in this paper and its contribution to the field and look forward to having it published.

The following document contains our response to the reviewers' comments. We very much hope the revised manuscript has addressed all the comments and concerns.

Thank you for your time.

Best, Kimberly L. Miller Douglas Jerolmack  
The work of Miller and Jerolmack entitled 'Controls on the rates and products of particle attrition by bed-load collisions' deals with earth surface dynamics, by taking into account fracture mechanics, leading to a very interesting and well treated multi-disciplinary approach.

The authors realized well controlled experiments of particle impact and attrition and clever data analysis, as well as precise size measurements of attrition products, allowing them to get their main experimental results: impact erosion can be treated 'as brittle fracture in the purely elastic regime'. Additionally, their fine observations of chipped particles allow them to support that 'the common fatigue failure model is inappropriate', but 'propose that Hertzian fracture is the dominant mechanism'. Again, materials mechanics appear surprisingly as a relevant tool for bedrock erosion, sand production, bed-load transport, ... The authors also consider the limitations of the methods and take time to explain them to the readers, that is really appreciated.

The whole work is realized rigorously. High numbers of different experiments are done to ensure good statistics (450 collisions to test the randomness of the grain rotation, 50 to 10 000 collisions, 20 000 collisions, . . .), that is really appreciated. I also really appreciate the analysis of experimental data by using dimensional analysis and knowledge from elasto-plasticity, as well as the desire of the authors to use 'physically-meaningful quantities'.

For all these reasons, I agree the publication of the paper. However, I have some

suggestions reported in sequence in the pdf file.

»Thank you for taking the time and effort to apply your expertise to reviewing our manuscript. We have addressed your comments below.

Supplement:

The work of Miller and Jerolmack entitled Âń Controls on the rates and products of particle attrition by bed-load collisions Âż deals with earth surface dynamics, by taking into account fracture mechanics, leading to a very interesting and well treated multi-disciplinary approach.

The authors realized well controlled experiments of particle impact and attrition and clever data analysis, as well as precise size measurements of attrition products, allowing them to get their main experimental results: impact erosion can be treated Âń as brittle fracture in the purely elastic regime Âż. Additionally, their fine observations of chipped particles allow them to support that Âń the common fatigue failure model is inappropriate Âż, but Âń propose that Hertzian fracture is the dominant mechanism Âż. Again, materials mechanics appear surprisingly as a relevant tool for bedrock erosion, sand production, bed-load transport, ... The authors also consider the limitations of the methods and take time to explain them to the readers, that is really appreciated.

The whole work is realized rigorously. High numbers of different experiments are done to ensure good statistics (450 collisions to test the randomness of the grain rotation, 50 to 10 000 collisions, 20 000 collisions, ...), that is really appreciated. I also really appreciate the analysis of experimental data by using dimensional analysis and knowledge from elasto-plasticity, as well as the desire of the authors to use Âń physically-meaningful quantities Âż.

For all these reasons, I agree the publication of the paper. However, I have some suggestions below that are reported in sequence.

Maybe it would be interesting to say a word about the case of an impact of a 'pebble'

with a granular material (with or without cohesion), the last one as a model of some riverbeds. Such non-consolidated (discrete) materials would be another class of materials, where there is not any true fracture, but where mass loss can occur. I know the papers from Beladjine et al 2007 (PRE), where they found a relation between the mass loss Nej, the effective restitution coefficient e function of the impact angle, and the Froude number. Are the experimental data or the current knowledge enough conclusive to be compared to your Eq (2) page 2: Delta M = A Delta E? However, maybe these experiments are not realized Âń under conditions relevant for bed-load transport Âż?

» We are quite familiar with the work cited by the reviewer, and related articles, on the controls of energy and impact angles on ejection of grains in aeolian transport. Indeed, we found a similar relation between impact energy, and cumulative energy of ejected particles, for bed-load transport underwater (Lee and Jerolmack, ESurf, 2018) – where we connected our result to this previous work. But this "mass loss", in terms of the number of particles ejected in a collision, is a very different process from mass loss of the impactor itself by collision and fracture. We want to be sure not to conflate the two. That said, it is certainly worthwhile to add some discussion about if/how collisions in our experiment are representative of natural collisions – or, what the important differences are. From this perspective, the reference above and others can be used to suggest how (typically) oblique collisions of natural pebbles, and other kinds of motion including rotation, may make the energy transfer different from what we do in our experiments. This is also a request from Reviewer 2. We have added a paragraph in the Methods section: "A note of caution is in order regarding the geometry and kinematics of our binary collisions, compared to the situation of bed-load transport. Fluvial pebbles impact the bed at shallow angles, typically on the order of $\theta \sim 10^o$. Such shallow angles reduce the bed-normal collision velocity by a factor $sin \theta$ \cite{sklar04, beladjine2007collision, larimer2021flume}, and proportionately reduce the mass lost per impact \cite{larimer2021flume, francioli2014characterizing}. Bed-load particles may also rotate \cite{francis1973experiments}, adding an additional

tangential velocity component to collisions. The effect of this rotation on mass attrition, however, has not been studied. Moreover, it has been suggested that rotation is small compared to the magnitudes of horizontal and vertical velocities associated with saltation \cite{nino1998using}. The rounding of fluvial pebbles in nature indicates that bed-normal chipping, rather than tangential (sliding) abrasion, is the dominant attrition mechanism under saltation \cite{novak2018universal}. The usual assumption in bed-load attrition studies is that collision energy is determined by the bed-normal component of saltation velocity, which is roughly the terminal fall velocity of the particle \cite{sklar04}. Despite the simplified collision scenario of our experiments, collision velocities are comparable in magnitude to computed terminal fall velocities for similar-sized particles in water. We expect then that experiments can be used to examine material and energy controls on mass loss, but that observed trends will include an empirical prefactor that is related to the specific details of our configuration.

References added: Larimer, J. E., Yager, E. M., Yanites, B. J., & Witsil, A. J. C. (2021). Flume experiments on the erosive energy of bed load impacts on rough and planar beds. Journal of Geophysical Research: Earth Surface, 126, e2020JF005834. https://doi.org/10.1029/2020JF005834. Beladjine, Djaoued, et al. "Collision process between an incident bead and a three-dimensional granular packing." Physical Review E 75.6 (2007): 061305. Francis, J. R. D. "Experiments on the motion of solitary grains along the bed of a water-stream." Proceedings of the Royal Society of London. A. Mathematical and Physical Sciences 332.1591 (1973): 443-471. Niño, Yarko, and Marcelo García. "Using Lagrangian particle saltation observations for bedload sediment transport modelling." Hydrological Processes 12.8 (1998): 1197-1218. Francioli, Daniel, et al. "Characterizing attrition of rock under incremental low-energy impacts." XXVII International Mineral Processing Congress-IMPC 2014: Conference Proceedings. Vol. 1. 2014.

I would like to see some references about papers from the mechanical community about brittle fracture in the purely elastic regime, if relevant for your work.

» This is more of a difference in semantics than a difference in mechanics. The most important point, mechanically, is that in Hertzian fracture the elastic wave fractures rock where the strain achieves a critical value. This kind of impulsive elastic wave in collision is different from a quasi-static loading – where stress increases linearly in the elastic regime and then becomes plastic right before failure. What we are trying to make a distinction about is that Hertzian fracture is the relevant mechanism for collisions. We have modified the text to make this distinction clearer.

Main text

1) Introduction. When I read Âń attrition rate Âż for the first time in your paper (line 52, page 2) and later (line 259 page 9), I am wondering to which definition you refer to: a time derivative of mass or a derivative of the mass according to the impact energy? I guess it is the derivative of the mass according to the impact energy, however it is not obvious in general.

»Yes, we are referring to the mass loss by impact energy. We have added clarification to both locations in the manuscript.

Maybe you should refer to the Charpy impact test, that look like your experimental set-up, even if boundary conditions are not the same. With such a test, do you think that the measurement of absorbed and/or released energy during the impacts and rebounds would be possible and interesting for your open questions? I think this may be related to one of your conclusion l. 440 p. 14 Âń We hypothesize that this coefficient is primarily controlled by the details of the collision process, which determine how much impact energy contributes to damage as opposed to friction or rebound of the target. Âż

» We have added reference to the Charpy test, for readers that may be familiar with that technique: "Our experiment bears some similarity to the Charpy impact test \cite{leis2013charpy} — a standard technique for measuring the energy absorbed by a material (typically metal) in producing fracture — but has modified boundary conditions

and geometry to better approximate binary bed-load collisions." In principle it would be possible to measure the rebound speed of the impactor, and the target, which could provide some additional information. But we cannot with the current setup find a way to independently measure energy loss due to friction; therefore, we would still be unable to directly determine the energy absorbed by the material that goes in to creating fractures. We kept our focus on relating mass loss to the impact energy for comparison to bed-load impact attrition studies. Also, it became impractical to measure rebounds and motions for tens of thousands of collisions with the high-speed camera. ADDED reference: Leis, Brian N. "The Charpy impact test and its applications." Journal of Pipeline Engineering 12.3 (2013): 183-198.

2) Methods. 2-1) Dimensional analysis is really appreciable, I think that it may be very useful. However, whereas it is clear for the brittle/purely elastic regime, I find it a bit less clearly written for the semi-brittle case. I do not know if you should present the ratio H/Kc as the third dimensionless group and/or you should explain why D, the sample diameter appears as an input parameter in the semi-brittle regime but not in the brittle regime (localized plastic deformations at the surface have to extend to the bulk, on a size D?)?

»We have clarified this by introducing the material properties earlier as important properties in semi-brittle deformation.

At this end, it is not crucial, because your experiments are well described by Ab, and not by As. But, this lacks. Also, it is not clear what are the units of parameters appearing in As: what is the unit of H (and Kc), so that As have the same units as Delta M/ Delta E and Ab? Nowhere in the paper, values of H are given, whereas As is computed?

»We have added values of As and Ab to table 1; these quantities have the units of attrition rate (s2/m2). Tensile strength, Youngs Modulus, and Hardness have units of stress (Pa), but Fracture toughness has units of stress times sq. of length (Pa m^0.5), therefore the length of the sample is included in the dimensional analysis to obtain the

correct units.

2-2) How do we know typical impact velocities or energies involved in bed-load transport to state that the values reported here in the experiments are Âń under conditions relevant for bed-load transport Âż?

» We compared to values published in the literature. We have added references to this line.

l.176: Why the area coming into play in the expression of the tensile strength is pi l D /2 instead of pi Dˆ2 / 4?

»The tensile strength as measured using the Brazilian tensile test is calculated using the expression noted in the manuscript because the stress is equally spread over the length of a cylindrical sample.

l. 208: The number As is called here the Attrition Number for the first time, you should have introduce this name for As and Ab, when these latter are introduced page 4.

»We have added references to earlier in the manuscript for clarification.

3) Results l. 252: In the sentence Âń Mass loss curves for all experiments are in good agreement with each other, and with a single linear trend (Fig. 6). Âż, the group Âń a single linear trend Âż would suggest that the slope is the same, that is not true. Maybe you should add Âń with different slopes Âż or something like that.

»Figure 6 shows a collapse of the different experimental runs for different samples of the same material – brick. In other words, this plot demonstrates the variability of the measurements. If we compare this plot, which contains only one lithology, to that of figure 5, which contain all lithologies tested, we see that the brick fall onto a single linear trend. We show this data to demonstrate that the results for abrasion number are consistent over a range of impact energies (different initial sized samples).

I am wondering why you choose to write your Eq (7) page 9 as M/M0 = k E + b, instead

of writing k as 1/Es; this would be more direct and this would avoid introducing two variables (k and Es) instead of only one.

»We chose to write the equation the way we did because we were calculating $k$ directly from the regressions of the data set.

l. 260, I would make a remind here on what is A: the Attrition number.

»We have added clarification.

l. 265: Âń The brittle Attrition Number Ab is plotted against long term attrition rates dM/dE (Fig. 7b) and demonstrate good correlation Âż, it should be added here that there is some scattering. Fig 7b: To which materials correspond the plotted data?

»We have added clarification that this plotted data is for all samples and that there is some scatter in the data.

Maybe, it should be written somewhere that Ab is between 0 and 1 and As between 0 and 2, thus are of the same order of magnitude. Say something on the precision or uncertainty of Ab (and As). Say something about the slope of the order of 10ˆ-5. As it is very far from 1, it should mean that some physical understanding still lacks, that will need further investigations in the future.

»We have added clarification about the order of magnitude of the parameter As, as well as information about uncertainty in the material properties. The section now reads: "The brittle attrition number $A_b$ is plotted against long term attrition rates $M/E$ for all samples(Fig. 7b) and demonstrate good correlation with some scatter, likely due to uncertainty in material property measurements, indicating that the brittle attrition number incorporates appropriate material properties to describe the long-term attrition of different lithologies. Although same order of magnitude as $A_b$, the semi-brittle attrition number $A_s$ varies widely. . ."

l. 272: Âń The parameter b is related to the initial mass of each particle, with an average value of b = M/M0 = 0.0018 and is approximately constant for all experiments

(Fig. 7c). Âż I don't see that b is approximately constant for all experiments in Fig. 7c.

»We have clarified this sentence. The value of b is related to the amount of rapid abrasion that occurs during the initial phase of the experiments when the angled corners are rounded and is therefore related not to the "initial mass" but the "initial mass loss".

l. 276: Âń results show that the former tracks the latter, and becomes approximately constant when rock mass $M/M_0 = 0.0018$ Âż I would write instead Âń $M/M_0 >= 0.0018$ Âż

»We have updated this equation to reflect your suggestion.

l.277: Âń This value is the same as b, Âż The value of b is not shown (in a Figure) or given (in Table 1). This lacks.

»We have added references to the plot (figure 7a) from which the value is taken from.

l. 281: Why introducing a new symbol C2, since it is b? I would change the sense of presentation of Eq.(8) as: $M/M_0 = C_1 A_b E + C_2 = C_1 \rho Y E /\sigma^2 + C_2$

»We introduce the new symbol C2 to indicate that there are two constants based on material and physical properties that control attrition besides the input of impact energy.

Also, it seems to me that there is a slight approximation because k is not equal to C1 Ab but is equal to C1 (Ab - 0.2) as can be seen in Fig. 7b. So, the term - 0.2 C1 E lacks in Eq(8). You should say a word about this.

» We apply equation 8 to the second phase of attrition once the high curvature sharp edges are rounded. We have clarified this by rewriting the following sentence: "Linear fits were then made to the second slower phase of all mass loss curves, resulting in the relation: $M/M\_0 = kE + b$."

l. 285: In accordance with Eqs (7) and (8), M should be divided by M0 in Eq (10). As a consequence, the sentence in l. 286 should be added by Âń and divided by M0 Âż.

»We have made the change by adding M0 to the equation.

l. 299: Instead of Âń diverge Âż, I would use Âń differ Âż to avoid suggesting wrongly tend towards infinity.

»We have clarified this sentence.

l. 312: Add the reference to Eq (6) in the sentence Âń We then solve for the best fit power law to all data points. Âż

»We have added the reference.

4) Discussion l. 322: I would add that the sentence Âń However, rocks achieve the secondary linear mass loss curve quickly while their shapes are still very close to cuboids. concern rocks observed here Âż or something like that.

»We have added to the sentence for clarity.

L. 324: I would begin a new paragraph to highlight the equation M/(M0 E) = k = C1 A and I would change the first symbol = by the symbol $\simeq$.

»We have changed the symbol to $\simeq$.

l. 329: Âń It appears our data are reasonably well described by Ab and not by As, indicating that material failure may be considered to be in the brittle regime. Âż This interesting conclusion stem from the correlation of your measurements with the brittle Attrition number AB: could we have inferred it without your measurements, but from the values of elasto-properties (or from the comparison of the values of Ab and As)? If not, which data (impact stress?, impact deformation? ...) should we have access to so that it becomes possible?

»The attrition numbers, both As and Ab are based on the physical properties of the material as discussed in the introduction of the manuscript. The brittle attrition number describes brittle fracture, whereas the semi brittle number describes more elastic-plastic brittle deformation. The mass loss data in this study is correlated to the brittle

attrition number indicating that failure is due to brittle fracture.

l. 331: When using Ab here, I would recall its relation with Ab=rho Y / sigmaˆ2.

» We have added the relations for both Ab and As in this sentence.

l. 346: I would refer to Fig. 4 and 8 in the sentence Âń The SEM images of sectioned rocks show a zone of damage accumulation in a shallow region below the surface. Âż

»We have added the figure references.

l. 367: attrtition-product -> attrition-product

»Spelling has been corrected.

l. 395: Âń In the limit where k = 0, the brittle Attrition Number, Ab, does not likewise approach zero, but instead is associated with Ab = 0.25. Âż It is not clear to me where this rationale comes from? From Fig 7b, I can read that Ab(dM/dE=0) = 0.25. But this not the value your reported. Explain better, please. And what is the unit of Ab? Here it appears that it is without unit. But in Eq (4), Ab has the unit of time square / meter square. Clarify it please.

»We have added clarification by adding both units and figure reference from which the observation is made.

l. 402: Âń However, the pebbles from the field were all at least 4 times larger than those used in the laboratory, while estimated collision velocities were comparable. Âż Does it mean that your experiments are done at the same velocities as in the field, but at smaller energy? Do you think that normalizing the mass loss by the initial mass allows suggesting that your results to become independent on M0? Can you still write that you are in the same conditions as in the field (Âń under conditions relevant for bed-load transport Âż)?

»You are correct. In our study the impact velocities are the same but the size of grains is smaller than those of the comparison field study in Puerto Rico, however the size and

velocity of grains used in this experiment are observed in other natural settings. The data of different sized brick samples (fig 5), suggest that there is a linear relationship between impact energy and mass loss, so we feel confident that we can compare our results to the field site.

Figures Figure 1: I found it not clear at the beginning that grains were free to rotate, maybe your sketch Figure 1 should be completed with additional arrows to indicate the movements direction, and maybe some words should be added in the main text.

»Arrows are shown in figure 2b indicating that the grains can rotate freely. We have also added to the main text.

Figure 2: I would add x=40mm, 80mm, 120mm or add the word Âń square Âż to Âń rock Âż and one additional value x, so that it is clear that the distances x=0, 40, 80, 120mm correspond to the square rock corners.

»We have added clarification of the corner locations to the figure caption.

Figure 5: Both lin-lin and log-log plots are really appreciated. You should cite your Eq (7) p. 9 in your legend, when talking about Âń linear fits Âż.

» We have added the equation reference.

Figure 6: In your inset, you show average mass, we would like to see also some error bars describing the standard deviations of mass data. How your data shown in inset are compared to your Eq (7)?

» Figure 6 inset is showing how the initial size, and therefore the magnitude of the impact energy scales linearly with mass loss for the same material (brick). In the main plot, the slopes are all similar (eq7), but the inset is highlighting the effect of impact energy.

Figure 7a: In the legend, you should add the precision Âń for all samples Âż.

»We have added this phrase to the caption.

Figure 7b: To which materials correspond the plotted data? Maybe, it should be written somewhere in the text that Ab is between 0 and 1 and As between 0 and 2, thus are of the same order of magnitude. Say something on the precision or uncertainty of Ab (and As).

»We have added a legend denoting the material for each data point. We have added sentence about values of Ab and As; see response above.

Also, there is some scattering in the data: according to you, what is its origin?

» The scatter is seen in the 5 different brick samples, which may be due to the natural variability and uncertainty in measurements of attrition experiments and material properties.

Figure 7c: Write in your legend that data are for two samples: which ones?

»We have added this information.

Table 1: You shout add some properties here: Attrition numbers Ab and As, H and Kc? The tensile strength is referred to as sigma_t here, whereas in the whole text, one finds sigma, sigma_t, sigma_f, ... one may get lost. Can you harmonize it please?

»We have added material properties to table 1. Additionally, sigma_t and sigma were both used to denote the tensile strength of the material. We have gone through the manuscript and streamlined it to all be just sigma. Sigma_f on the other hand refers to the failure strength, which is different that the tensile strength. We have tried to clarify this is the manuscript.

Sometimes, figures or equations are mis-referred: p. 10 l. 288: Fig 5 b/c -> Fig 4 b/c Figure 8: In the legend, Eq (5) -> Eq (6). Figure 9: In the legend, Eq (3) -> Eq (6).

»These have been corrected.   Interactive comment on "Controls on the rates and products of particle attrition by bed-load collisions" by Kimberly Litwin Miller and Douglas Jerolmack

Sebastien Carretier (Referee) sebastien.carretier@get.omp.eu

The authors present a very interesting contribution on pebble attrition. Based on a large number of pebble impact experiments, the authors conclude that size reduction rate is proportional to energy and occurs mainly during elastic impacts rather than by damage, with the production of a fairly fine material whose grain size is distributed as a heavy-tailed power law. This paper is very well written and illustrated and the conclusions are supported by robust statistics over a large number of samples. In particular, Figure 7b illustrates the main conclusion. I have only minor remarks.

The first concerns the relationship between the experimental set-up and the transport of pebbles in a river. From what I understand (but I may be wrong) the shock is frontal in the experiments even if there is a certain degree of freedom in the pendulum movement. In rivers the pebbles can roll and rotate, with a tangential component in the shock. So I wonder if this difference may affect the transposition of the experimental results to natural rivers.

»» This is related to a point brought up by Reviewer 1. We have added a new paragraph on this in the Methods section (see response to Reviewer 1).

Looking at the pictures in thin sections, it seems that a damage zone forms well with a width that seems to be greater than the (potentially future) "grains" delimited by the fractures. This suggests that the formation of this band still partly controls the detachment of grains, and thus that damage plays a role. I recognize the hypothesis test on the basis of the Ab (brittle) and As (damage) parameters in Figure 7b, but I have difficulty in linking a purely brittle behaviour to this band which could be related to damage, although the authors recognize that more work is needed to verify that this band corresponds to a Hertzian fracture band.

» This comment is helpful. It shows our writing was unclear. We view Hertzian fracture, and damage, to be compatible. Indeed, the conceptual model that we have is that

the elastic shock fractures material for which the strain exceeds a critical value. This doesn't necessarily mean that the fractured materials disintegrate. In other words, it is likely that repeated Hertzian fracture creates a damaged skin dense with fractures, which then goes on to disintegrate from further collisions. To us, damage simply means loss of competence/strength but without disintegrating. The important distinction to us is only that this damage is confined to skin depth; it is not activating cracks deep in the bulk. We do concede some localized plastic deformation likely occurs (must occur?) at the base of impact zone – since this skin depth represents a yield surface, which is typically associated with plastic deformation. We do not have enough data to make such a distinction.

Line 58 should be (e.g., Shipway and Hutchings, 1993).

» This citation has been corrected.

Line 92. If deformation is "purely elastic" it would be reversible and thus without perennial cracks or fractures.

» Good point. What we meant to say is that it is related to an elastic shock wave the strains the material beyond a critical value, which then eventually causes (plastic) yielding. We have modified the text in SEVERAL places to make this more clear.

Line 355-358 A little more explanation of this difference here might help to better understand.

» We have actually removed text about this distinction – mostly because of other comments and questions from the reviewers. Our data do indicate that the brittle attrition number describes mass loss. And, the qualitative thin section images of cracking show that the chipping is not activating cracks/flaws in the bulk, but rather in a shallow skin. But, the "cyclic fatigue" point seemed to be a distraction for multiple reviewers, and we would rather not distract by adding discussion on a separate mechanism that is not part of this study.

Line 408 See also Jones and Humphrey, 1997.

»The citation has been added.

Best wishes, Sebastien Carretier   Interactive comment on "Controls on the rates and products of particle attrition by bed-load collisions" by Kimberly Litwin Miller and Douglas Jerolmack Jeffrey P Prancevic (Referee)

jeff.prancevic@gmail.com

Review of "Controls on the rates and products of particle attrition by bed-load collisions" by Kimberly Litwin Miller and Douglas Jerolmack

This manuscript presents the results of a fascinating set of experiments and provides semi-empirical predictions for both rates of sediment attrition and the size distributions of the attrition products. The experiments were nicely designed, and the measurements were detailed and complete (except in cases where certain material properties weren't possible to measure). The authors used a pair of pendulums to repeatedly collide two grains of the same size and rock type. Characteristic velocities (and kinetic energies) were measured with high-speed cameras, and masses were measured between sets of collisions. SEM imagery of thin sections of particles after the experiments were used to observe and measure the development of fractures. Material properties were measured with various standard methods.

The manuscript presents several interesting findings, but, from my perspective, the most important claim is that the rate of mass loss (as a function of impact energies) can be predicted from material properties and empirical constants measured in this study. However, this claim is not adequately demonstrated for several reasons that are outlined below. These issues should be addressed before the manuscript is accepted for publication. Otherwise, the manuscript is well-written, easy to follow, and full of cool observations. I do list several minor points by line number below.

»Thank you for taking the time to review the manuscript. We appreciate the feedback.

Issues with the prediction of the rate of mass loss

Equation (8) presents an elegant model of mass loss as a function of impact energy, material properties, and two empirical constantsâATC1 and C2. However, the observational basis for both C1 and C2 is shaky, based on the information presented in the manuscript. C1 is based on the best-fit curve between Ab (based on material properties) and the ratio of mass lost to cumulative energy, shown in Figure 7b. This plot shows a cloud of 8 data points in the middle with no strong relationship, and two outliers–one with high Ab and one with small Ab. There is no legend to identify these data points, but based on information in Table 1, the small-Ab data point represents the volcanoclastic cubes. The authors were unfortunately not able to measure the Young's modulus for the volcanoclastic samples because they specimens were too short, and they instead rely on values from the literature, which span an order of magnitude: Y is between 5 and 50 GPa. Therefore, possible values for the Brittle Attrition Number (Ab, the x-axis of Figure 7b) also span an order of magnitude. Strangely, based on the Figure 7b and the other material properties for the volcanoclastic rocks presented in Table 1, the authors used a value for Young's modulus that is outside the range found in the literature: Y = 4 GPa. The authors should double-check the values used in their calculations and be explicit about the material properties used to estimate Ab. For experiments where material properties are looked up from the literature, the manuscript should present a range of values of C1.

» We have added a legend to figure 7b. Furthermore, we have checked the values of the young's modulus and is it not outside those listed for values cited in the literature. We use the value of 35 GPa for the volcanoclastic samples, which is in the middle of the range listed. We have noted within the manuscript which properties were directly measured and which were collected from previous literature.

The issue with C2 is less important, particularly since the authors note that this early mass loss is mostly due to the cuboid shape of the particles and is likely unimportant for natural particles. Still, I was confused as to why Figure 7c only shows experimental

results from the sandstone and quartz diorite experiments, while the manuscript claims that the early mass-loss behavior is universal. If it's truly universal, it would be more compelling to show that behavior for all of the particles.

» Unfortunately, we only measured SHAPE for these two particles. The initial different behavior for MASS LOSS of all materials is similar. The shape measurements, for the two materials on which it was performed, show that the initial regime of different mass loss behavior corresponds to an initial regime of different shape evolution. We then infer that this is true for the other materials, but we did not directly measure them. While this was in the text, it appeared at the end of the section and therefore it was easy to miss. We have added a sentence in the beginning of the paragraph on shape to make this clear.

Finally, Equation 8 is calculated from several steps, not one regression, and the predictive ability of Equation 8 is not tested. This model should be compared against the data shown in Figure 5a to show how well it predicts mass loss. This isn't a true test of the model, since it's comparing it against the data used to create the model, but it's better than nothing.

» The equation 8 is presented, in a sense, as a summary of the relations that have been determined and demonstrated in the paper. We consider the data collapse demonstrated in Figure 5b to be the most fundamental result of the mass loss part of this paper. In other words, it demonstrates directly that the mass loss curves are controlled by only two parameters, the slope and intercept. We intentionally did not plot equation 8 against data because, as the reviewer points out, this wouldn't really be a test of the model since it would be comparing to data used to create the model. We consider this sufficient reason to object to making such a plot, because it could IMPLY that we consider this a model test. We believe the most important test is the one of linearity in Figure 5b, and the collapse of the data from two parameters that are shown to have at least SOME physical meaning (material and initial shape control for the slope and intercept, respectively). At equation 8, we have added a sentence saying that the data

collapse in figure 5b validates this equation.

Comments by line number

23. Consider replacing "it" with "attrition" to avoid confusion with "abrasion"

»We have clarified the sentence.

58-73. I'm not an expert in fracture mechanics, so much of this discussion here and in the discussion section (354-365) was difficult for me to follow. That said, my reading of this section is that it is commonly assumed that attrition processes occur by the progressive development of cracks through the entire particle, rather than local fractures (surface parallel or otherwise). This certainly isn't the conceptual model that I would normally assume (I would local fracture around the impact site to be important), but if that's what people do normally assume then it's very good to point that out. Right now this statement of what is "typically assumed" is supported by only one reference from 60 year ago.

» Some of the older mechanics models, which Sklar and Dietrich draw from, are implicitly built on a model for fracture by fatigue failure. That said, we have realized that introducing this contrast of chipping with fatigue failure is a bit of distraction – based on the reviews. So, we have basically removed discussion of the cyclic fatigue failure and simply described what we see in our experiments.

114-115. These sentences require the reader to differentiate between "significantly smaller" and "much smaller." Consider rewording.

»This sentence has been reworded for clarity.

262. Why are dM and dE used for cumulative values of mass loss and energy expenditure? Aren't M and E defined the same way? If there is a need to differentiate these values from M and E, consider using big delta, rather than the derivative.

» We have changed the text to clarify this. Indeed, the concept of dM/dE is important in

the sense that the mass loss rate should in principle be the derivative. But, as pointed out, in practice this was computed as (M-0)/(E-0) which is a finite difference. So we now write simply M/E.

313. Exponents should be negative (-2.5 and -2)

»This has been corrected.

367. "attrition" spelling

» We have corrected the spelling.

Table 1. Consider adding columns for total number of impacts, Ab, and As. Figure 7c. Show data from all experiments here

»We have added Ab and As to the table. For figure 7c, we can only show data for quartz diorite and sandstone samples, as data was not collected for all study specimens.

Figure 8b. Why are lengths not shown for the volcanoclastic rocks?

»The lengths were not measured for this lithology so was not included.

Please also note the supplement to this comment:
https://esurf.copernicus.org/preprints/esurf-2020-86/esurf-2020-86-AC1-supplement.pdf